# PHARMACOMATCH: EFFICIENT 3D PHARMACOPHORE SCREENING VIA NEURAL SUBGRAPH MATCHING

**Daniel Rose[1,2,3], Oliver Wieder[1,2]\*, Thomas Seidel[1,2] & Thierry Langer[1,2]**
[1]Department of Pharmaceutical Sciences, Division of Pharmaceutical Chemistry,
Faculty of Life Sciences, University of Vienna, 1090 Vienna, Austria
[2] Christian Doppler Laboratory for Molecular Informatics in the Biosciences,
Department of Pharmaceutical Sciences, University of Vienna, 1090 Vienna, Austria
[3]Vienna Doctoral School of Pharmaceutical, Nutritional and Sport Sciences,
University of Vienna, 1090 Vienna, Austria
*Email: {daniel.rose, oliver.wieder, thomas.seidel, thierry.langer}@univie.ac.at*

## ABSTRACT

The increasing size of screening libraries poses a significant challenge for the development of virtual screening methods for drug discovery, necessitating a re-evaluation of traditional approaches in the era of big data. Although 3D pharmacophore screening remains a prevalent technique, its application to very large datasets is limited by the computational cost associated with matching query pharmacophores to database molecules. In this study, we introduce PharmacoMatch, a novel contrastive learning approach based on neural subgraph matching. Our method reinterprets pharmacophore screening as an approximate subgraph matching problem and enables efficient querying of conformational databases by encoding query-target relationships in the embedding space. We conduct comprehensive investigations of the learned representations and evaluate PharmacoMatch as pre-screening tool in a zero-shot setting. We demonstrate significantly shorter runtimes and comparable performance metrics to existing solutions, providing a promising speed-up for screening very large datasets.

## 1 INTRODUCTION

The immense scale of the chemical space, covering over $10^{60}$ small organic molecules (Virshup et al., 2013), makes the identification of molecules that bind to a protein target particularly challenging. Virtual screening methods are important tools in computer-aided drug discovery, addressing this complexity and supporting medicinal chemists navigate large molecular databases in search of potential hit compounds (Sliwoski et al., 2014). Among these methods, an established approach is 3D pharmacophore screening. Pharmacophores represent an ensemble of steric and electronic molecular features that is necessary to ensure an optimal interaction with a specific biological target (Wermuth et al., 1998). A pharmacophore query can, for instance, be generated from the interaction profile of a ligand-protein complex and then aligned positionally with the three-dimensional conformations of compounds in a database (Wolber & Langer, 2005). These molecules are subsequently ranked based on their agreement with the pharmacophore query, with those exhibiting similar pharmacophoric patterns retrieved as potential hit compounds (Wolber et al., 2006).

An important development of the last years has been the emergence of make-on-demand libraries, such as Enamine REAL (Shivanyuk et al., 2007). These libraries contain billions of commercially available compounds that can be rapidly synthesized and are continuously expanding, driven by advances in synthetic accessibility (Llanos et al., 2019). Screening larger libraries increases the chances of identifying hits, but it also brings the trade-off of longer screening times. Scaling up 3D pharmacophore screening to handle billions of molecules is challenging due to the computational cost of pharmacophore alignment (Warr et al., 2022). Despite efforts to optimize these algorithms (Wolber et al., 2008; Permann et al., 2021) and the development of various pre-filtering techniques (Seidel et al., 2010), the alignment step remains a critical bottleneck.

---

*Corresponding author

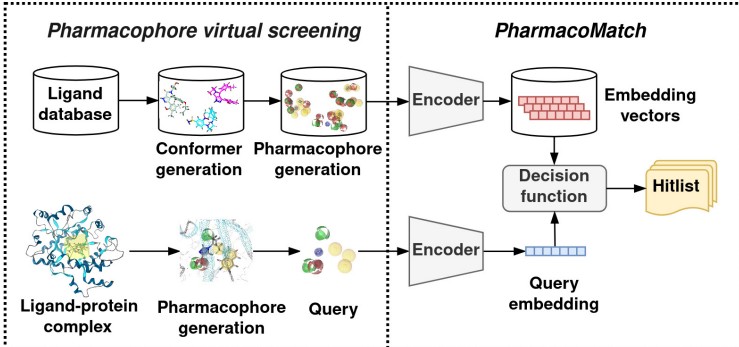

Figure 1: Overview of the *PharmacoMatch* workflow: Conformer and pharmacophore generation from ligands and query creation, for example from a ligand-protein complex, precede pharmacophore screening. The encoder model converts the screening database into embedding vectors, stored for later use. A hitlist is generated by comparing the query embedding with the database embeddings.

In this work, we propose to perform 3D pharmacophore screening by using learned representations. Specifically, our *PharmacoMatch* model employs a graph neural network (GNN) encoder to map 3D pharmacophores into an order embedding space (Ying et al., 2020), and predicts pharmacophore matching through vector comparisons. The embedding vectors for the screening database are computed once and then used to quickly generate a hitlist based on the query embedding (Figure 1). Our key contributions are as follows.

- We conceptualize pharmacophore matching as a representation learning problem, enabling the use of order embeddings for efficient and scalable pre-screening.

- We develop a GNN encoder that generates meaningful vector representations from 3D pharmacophores. The model is trained in a self-supervised manner on unlabeled data, employing a contrastive loss objective to capture partial ordering relationships between queries and targets in the learned embedding space. We design augmentation strategies specifically suited for the task of pharmacophore matching.

- We thoroughly analyze the learned embeddings and validate the practical utility of our approach as an effective pre-screening tool through experiments on virtual screening benchmark datasets.

We empirically demonstrate that our method is significantly faster than existing solutions, offering a practical advantage for screening billion-compound libraries. A promising direction for further improving efficiency is combining our approach with approximate retrieval techniques. Our work thus represents a key step towards vector databases for virtual screening. Additionally, our reformulation of pharmacophore screening as neural subgraph matching directly captures its subgraph matching nature. To the best of our knowledge, we are the first to introduce order embeddings in virtual screening, providing a novel perspective for the field. Furthermore, our use of self-supervised learning presents a promising strategy for addressing data scarcity in drug discovery. We hope that the combined insights from our study contribute to advancing representation learning for virtual screening applications.

## 2 RELATED WORK

**Pharmacophore alignment algorithms** Alignment algorithms compute a rigid-body transformation, the *pharmacophore alignment*, to match a query's pharmacophoric feature pattern to database ligands. A *scoring function* then evaluates the *pharmacophore matching* by considering both the number of matched features and their spatial proximity. The alignment is typically preceded by fast filtering methods that prune the search space based on feature types, feature point counts, and quick distance checks. Only molecules that pass these filters undergo the final, computationally

expensive 3D alignment step, which is usually performed by minimizing the root mean square deviation (RMSD) between pairs of pharmacophoric feature points (Dixon et al., 2006; Seidel et al., 2010). The algorithm by Wolber et al. (2006) creates smoothed histograms from the neighborhoods of pharmacophoric feature points for pair assignment using the Hungarian algorithm, followed by alignment with Kabsch's method (Kabsch, 1976). A recent implementation by Permann et al. (2021) improves on runtime and accuracy by using a search strategy that maximizes pairs of matching pharmacophoric feature points. Alternatively, shape-matching algorithms like ROCS (Hawkins et al., 2007) and Pharao (Taminau et al., 2008) model chemical features by Gaussian volumes, optimizing for volume overlap.

**Machine learning for virtual screening**    A common approach to using machine learning for virtual screening is to train models on measured bioactivity values. However, these models are constrained by the scarcity of experimental data, which is both costly and challenging to obtain (Li et al., 2021). Unsupervised training of target-agnostic models for virtual screening avoids dependence on labeled data, but remains relatively unexplored. DrugClip (Gao et al., 2023), which is not based on pharmacophores, approaches virtual screening as a similarity matching problem between protein pockets and molecules, using a multi-modal learning approach where a protein and a molecule encoder create a shared embedding space for virtual screening. Sellner et al. (2023) used the Schrödinger pharmacophore shape-screening score to train a transformer model on pharmacophore similarity, which is a different objective than pharmacophore matching. PharmacoNet (Seo & Kim, 2023) is a pre-screening tool that uses instance segmentation for pharmacophore generation in protein binding sites and a graph-matching algorithm for binding pose estimation. They employ deep learning for pharmacophore modeling, but not for the alignment nor matching.

## 3    PRELIMINARIES

**Pharmacophore representation**    In this work, we treat 3D pharmacophores as attributed point clouds (Mahé et al., 2006; Kriege & Mutzel, 2012). A pharmacophore $P$ can be represented by a set of pharmacophoric feature points $P = \{(\mathbf{r}_i, d_i) \in \mathbb{R}^3 \times \mathcal{D}\}_i$ with the Cartesian coordinates $\mathbf{r}_i$ and the descriptor $d_i$ of the pharmacophoric feature point $p_i$. The descriptor set $\mathcal{D}$ covers the following *pharmacophoric feature types*: hydrogen bond donors (HBD) and acceptors (HBA), halogen bond donors (XBD), positive (PI) and negative electrostatic interaction sites (NI), hydrophobic interaction sites (H), and aromatic moieties (AR). Directed feature types like HBD and HBA can be associated with a vector component, but for simplicity, we will omit this information in our study. We further denote the set of pair-wise distances between feature points as $\mathcal{R} = \{\|\mathbf{r}_i - \mathbf{r}_j\|_2 \mid 1 \leq i, j \leq |P|\}$. The pharmacophore $P$ can be represented as a complete graph $G(P) = (V_P, E_P, \lambda_P)$, where $V_P = \{v_1, ..., v_{|P|}\}$ denotes the set of nodes with node attributes $\lambda_P(v_i) = l_i$, and $E_P = V_P \times V_P$ denotes the set of edges, with the edge attribute of $e_{ij}$ defined as the pair-wise Euclidean distance $\lambda_P(e_{ij}) = \|\mathbf{r}_i - \mathbf{r}_j\|_2$ between the positions of nodes $i$ and $j$. The edges are undirected, edge $e_{ij}$ can be identified with edge $e_{ji}$. The label set $\mathcal{L} = \mathcal{D} \cup \mathcal{R}$ is the union of the feature descriptor set and the set of pair-wise distances, and $\lambda_P$ represents a labelling function $\lambda : V \cup E \to \mathcal{L}$ that assigns a label to the corresponding vertex $v$ or edge $e$. This representation is invariant to translation and rotation.

**Subgraph matching**    Two graphs $G_1 = (V_1, E_1, \lambda_1)$ and $G_2 = (V_2, E_2, \lambda_2)$ are *isomorphic*, denoted by $G_1 \simeq G_2$, if there exists an edge-preserving bijection $f : V_1 \to V_2$ such that $\forall (u, v) \in E_1 : (f(u), f(v)) \in E_2$. Additionally, we require the preservation of node and edge labels, such that $\forall v \in V_1 : \lambda_1(v) = \lambda_2(f(v))$, and $\forall (u, v) \in E_1 : \lambda_1((u, v)) = \lambda_2((f(u), f(v)))$. Let $G_Q = (V_Q, E_Q, \lambda_Q)$ be a query graph, $G_T = (V_T, E_T, \lambda_T)$ a larger target graph, and $G_H = (V_H, E_H, \lambda_H)$ a subgraph of $G_T$ such that $V_H \subseteq V_T$, and $E_H \subseteq E_T$. The objective of *subgraph matching* is to decide, whether $G_Q$ is *subgraph isomorphic* to $G_T$, denoted by $G_Q \lesssim G_T$, requiring the existence of a non-empty set of subgraphs $\mathcal{H} = \{G_H \mid G_H \simeq G_Q\}$ that are isomorphic to $G_Q$.

**Pharmacophore matching**    In its most general setting, *pharmacophore matching* seeks to match all pharmacophoric feature points of a query pharmacophore $P_Q$ with the corresponding feature points of a larger target pharmacophore $P_T$. Let $P_H \subseteq P_T$ denote a subset of the feature points of $P_T$. Then $P_Q$ matches $P_T$ *after alignment* if there exists a bijection $g : P_Q \to P_H$ such that $\forall i \in P_Q : d_i = d_{g(i)}$ and $\|\mathbf{r}_i - \mathbf{r}_{g(i)}\|_2 < r_T$, where $r_T$ is the radius of a tolerance sphere. It

is thereby sufficient that query feature points are mapped into the tolerance sphere of their target counterpart. For simplicity, we assume the same tolerance radii among all pharmacophoric feature points. The ultimate goal of pharmacophore matching is to retrieve molecules from a database. A matching pharmacophore is always linked to a corresponding ligand molecule *via* a look-up table.

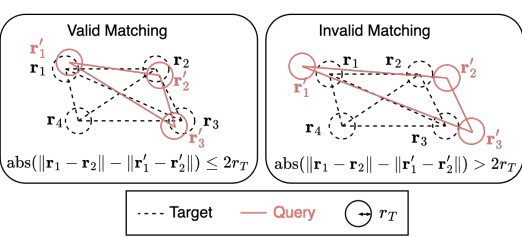

When represented as graphs $G_Q = G(P_Q)$, $G_H = G(P_H)$, and $G_T = G(P_T)$, this task boils down to the node-induced subgraph matching of a query pharmacophore graph $G_Q$ to a target pharmacophore graph $G_T$. The tolerance sphere, however, weakens the requirement on edge label matching. An approximate matching $\lambda_Q((u, v)) \approx \lambda_H((f(u), f(v)))$ is sufficient if the difference between $\lambda_Q((u, v))$ and $\lambda_H((f(v), f(u)))$ is less than $2r_T$, where $r_T$ represents the tolerance radius of each pharmacophoric feature point. This ensures that the query points fall within the tolerance spheres of the target points (compare Figure 2). Our problem formulation of pharmacophore matching relies on relative distances instead of the absolute positioning of pharmacophoric features

Figure 2: Illustration of the pharmacophore matching objective: The aim is to match the pharmacophoric points of a query with the corresponding points of a target pharmacophore such that the query points fall within the tolerance sphere of the target points, with a tolerance radius $r_T$.

and is therefore *independent of prior alignment*.

## 4 METHODOLOGY

**Overview**  In the following we introduce PharmacoMatch, a novel contrastive learning framework with the aim to encode *query-target relationships* of 3D pharmacophores into an embedding space. We propose to train a GNN encoder model in a *self-supervised* fashion, as illustrated in Figure 3. Our model is trained on approximately 1.2 million *unlabeled small molecules* from the ChEMBL database (Davies et al., 2015; Zdrazil et al., 2023) and learns pharmacophore matching solely from *augmented examples*, comparing positive and negative pairs of query and target pharmacophore graphs, while optimizing an *order embedding loss* to extract relevant matching patterns.

**Unlabeled data for contrastive training**  To span the *pharmaceutical compound space*, we download a set of drug-like molecules sourced from the ChEMBL (2024) website in the form of Simplified Molecular Input Line Entry System (SMILES) strings (Weininger, 1988) and curate an unlabeled dataset using the open-source Chemical Data Processing Toolkit (CDPKit) (Seidel, 2024) (see Appendix A.1 for details). After an initial data clean-up, which includes the removal of solvents and counter ions, adjustment of protonation states to a physiological pH, and elimination of duplicate structures, the dataset contains approximately 1.2 million small molecules. To ensure a *zero-shot setting* in our validation experiments, we remove all molecules from the training data that also appear in the test sets. Finally, we generate a low-energy 3D conformation and the corresponding pharmacophore for each ligand.

**Model input**  We represent the node labels $\{\lambda_P(v_1), ..., \lambda_P(v_{|P|})\}$ of a given pharmacophore graph $G(P) = (V_P, E_P, \lambda_P)$ as *one-hot-encoded* (OHE) feature vectors $\mathbf{h} = (\mathbf{h}_1, ..., \mathbf{h}_{|P|})$. We employ a *distance encoding* to represent pair-wise distances, which was inspired by the SchNet architecture (Schütt et al., 2018). The edge attributes of edge $e_{uv}$ are derived from the edge label $\lambda_P(e_{uv})$ and represented by a radial basis function $\mathbf{e}_k(\mathbf{r}_u - \mathbf{r}_v) = \exp(-\beta(\|\mathbf{r}_u - \mathbf{r}_v\|_2 - \mu_k)^2)$, where centers $\mu_k$ were taken from a uniform grid of $K$ points between zero and the distance cutoff at 10 Å, and the smoothing factor $\beta$ represents a hyperparameter. To this end, the pharmacophore $P$ is represented by a data point $\mathbf{x} = [\mathbf{h}, \mathbf{e}]$ which is a tuple of the feature matrix $\mathbf{h} \in \mathbb{R}^{|P| \times |\mathcal{D}|}$ and the distance-encodings $\mathbf{e} \in \mathbb{R}^{(|P| \times |P|) \times K}$.

**GNN encoder**  The encoder input is the pharmacophore graph representation $\mathbf{x} = [\mathbf{h}, \mathbf{e}]$, with the feature matrix $\mathbf{h}$ and the edge attributes $\mathbf{e}$. Node feature embeddings are generated by initially passing the OHE feature matrix through a single dense layer without an activation function. We

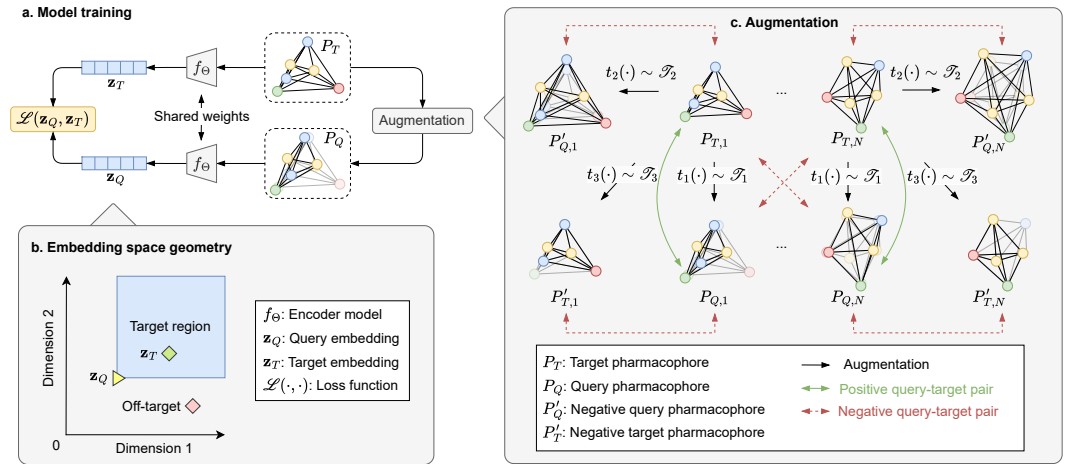

Figure 3: (a) The encoder model learns an order embedding space by comparing augmented pharmacophores. (b) Illustration of the embedding space, where pharmacophores matching a query are positioned to the upper right. (c) Augmentation strategies for model training involve generating positive and negative query-target pairs on-the-fly by combining node deletion with varying degrees of node displacement. Negative pairs are also created by shuffling the batch, mapping query pharmacophores to random target pharmacophores.

then update the node representations through *message passing* using the *edge-conditioned convolution operator* (NNConv) by Gilmer et al. (2017); Simonovsky & Komodakis (2017), which was originally designed for representation learning on point clouds and 3D molecules, to aggregate distance information into the learned node representations (see Appendix A.3 for details). We connect successive convolutional layers using DenseNet-style skip connections (Huang et al., 2017). Graph-level read-out is achieved by *additive pooling* of the updated feature matrix $\mathbf{h} \in \mathbb{R}^{|P| \times m}$ into a graph representation $\mathbf{q} \in \mathbb{R}^m$, which is then projected to the final output embedding $\mathbf{z} \in \mathbb{R}_+^D$ by a multi-layer perceptron. The employed loss function requires to map the final representation to the *non-negative real number space*. We accomplish this by using the absolute values of the learnable weights for the last linear transformation immediately after the final ReLU unit (see Appendix A.4 for details).

**Loss function**   In order to encode query-target relationships of pharmacophores into the embedding space, we employ the loss function by Ying et al. (2020). The key insight is that subgraph relationships can be effectively encoded in the geometry of an order embedding space through a partial ordering of the corresponding vector embeddings. Let $\mathbf{z}_Q$ the embedding of graph $G_Q$, $\mathbf{z}_T$ the embedding of graph $G_T$, and $f_\Theta : \mathcal{G} \to \mathbb{R}_+^D$ a GNN encoder to map pharmacophore graphs $\mathcal{G}$ to embedding vectors $\mathbf{z} \in \mathbb{R}_+^D$. The partial ordering $\mathbf{z}_Q \preceq \mathbf{z}_T$ reflects, whether $G_Q$ is subgraph isomorphic to $G_T$: $\mathbf{z}_Q[i] \leq \mathbf{z}_T[i], \ \forall i \in \{1, ..., D\}$ iff $G_Q \lesssim G_T$. The following max-margin objective can be used to train the GNN encoder $f_\Theta$ on this relation:

$$\mathcal{L}(\mathbf{z}_Q, \mathbf{z}_T) = \sum_{(\mathbf{z}_Q, \mathbf{z}_T) \in Pos} E(\mathbf{z}_Q, \mathbf{z}_T) + \sum_{(\mathbf{z}_Q, \mathbf{z}_T) \in Neg} \max\{0, \alpha - E(\mathbf{z}_Q, \mathbf{z}_T)\} \qquad (1)$$

The penalty function $E : \mathbb{R}_+^D \times \mathbb{R}_+^D \to \mathbb{R}_+$ reflects violation of the partial ordering on the embedding vector pair: $E(\mathbf{z}_Q, \mathbf{z}_T) = \|\max\{\mathbf{0}, \mathbf{z}_Q - \mathbf{z}_T\}\|_2^2$. $Pos$ is the set of positive pairs per batch, these are pairs of query $\mathbf{z}_Q$ and target graph embedding $\mathbf{z}_T$ with a subgraph-supergraph relationship, and $Neg$ is the set of negative examples, these are pairs of query and target embedding vectors that violate this relationship. The positive and negative pairs are generated on-the-fly *via* augmentation during training.

**Augmentation module**  The PharmacoMatch model correlates the matching of a query and a target pharmacophore with the partial ordering of their vector representations. Positive pairs represent successful matchings, while negative pairs serve as counter examples. In order to create these pairs from unlabeled training data, we define three families of augmentations $\mathcal{T}$, which are composed of *random point deletions* and *positional point displacements*. For positive pairs, valid queries are created by randomly deleting some nodes from a pharmacophore $P$, leaving at least three, and displacing the remaining nodes within a tolerance sphere of radius $r_T$. This augmentation, denoted as $t_1(\cdot) \sim \mathcal{T}_1$, produces the positive pair $(t_1(P), P)$. Negative pairs highlight examples of unsuccessful matching, using three strategies to capture different undesired outcomes. The first strategy introduces positional mismatches by displacing the pharmacophoric feature points of $P$ to the boundary of the tolerance sphere without deleting any points. Specifically, we compute the mean position of the points, $\mu = \frac{1}{|P|} \sum_{i=1}^{|P|} \mathbf{r}_i$, and shift each point $p_i$ by $r_T$ along the direction $\mathbf{r}_i - \mu$. This approach avoids the unintended creation of positive pairs, which can occur with random sampling, and ensures that displacements do not cancel out. The resulting augmentation, denoted $t_2(\cdot) \sim \mathcal{T}_2$, generates the negative query-target pair $(t_2(P), P)$. This strategy demonstrated better model performance than random positional displacement. Our second strategy teaches the model that every pharmacophoric feature point in the query should correspond to a point in the target. This is achieved by deleting some target nodes, using an augmentation operator $t_3(\cdot) \sim \mathcal{T}_3$, where $\mathcal{T}_3$ involves node deletion without displacement. As a result, the query in the pair $(t_1(P), t_3(P))$ only partially matches its target. With the third strategy, we train the model to avoid matching queries with targets that are significantly different. This approach involves randomly mapping queries $t_1(P_i)$ to the incorrect targets $P_j$, where $i \neq j$ (for more details, see Appendix A.2).

**Model details & design choices**  Our GNN encoder model is implemented with three convolutional layers with an output dimension of 64. The MLP has a depth of three dense layers with a hidden dimension of 1024 and an output dimension of 512. The final model was trained for 500 epochs using an Adam (Kingma, 2014) optimizer with a learning rate of $10^{-3}$. The margin of the best performing model was set to $\alpha = 100$. The default tolerance radius $r_T$ in CDPKit's pharmacophore screening is set to 1.5 Å, and we use the same value for the node displacement during model training to ensure consistency with the alignment algorithm in subsequent evaluations. We design a curriculum learning strategy for learning on pharmacophore graphs, detailed in Appendix A.5, along with details on model training and hyperparameter optimization. Systematic ablation studies of model architecture components and the impact of different augmentation strategies, model size, and embedding dimension on model performance are provided in Appendix A.5.

**Decision function for model inference**  We use the trained GNN encoder $f_\Theta$ to precompute vector embeddings $\mathbf{z}_T$ of the database pharmacophores. These are queried with the pharmacophore embedding $\mathbf{z}_Q$ by verification of the partial ordering constraint through penalty function $E(\cdot, \cdot)$ (Equation 1), which shall not exceed a threshold $t$. This leads to decision function $g : \mathbb{R}_+^D \times \mathbb{R}_+^D \to \{0, 1\}$:

$$g(\mathbf{z}_Q, \mathbf{z}_T) = \begin{cases} 1 & \text{iff } E(\mathbf{z}_Q, \mathbf{z}_T) < t \\ 0 & \text{otherwise} \end{cases} \tag{2}$$

evaluating 1, if the partial ordering on $\mathbf{z}_Q$ and $\mathbf{z}_T$ reflects a pharmacophore matching, and 0 otherwise. In the following, we refer to equation (2) as matching decision function. In practice, we recommend a decision threshold of $t = 6500$, determined from our benchmark experiments.

## 5 Experiments

We designed the embeddings to reflect the type and relative positioning of pharmacophoric feature points. Comparison of embedding vectors *via* the matching prediction function should emulate the matching of the underlying pharmacophores. To get a better understanding of the encoder's latent space, we investigate these properties as follows:

1. **Pharmacophoric feature point perception**: We investigate the learned embedding space qualitatively through dimensionality reduction.

2. **Positional perception**: We investigate the influence of positional changes on the output of the matching decision function.

3. **Virtual screening performance**: The performance of our model is evaluated using ten DUD-E (Mysinger et al., 2012) targets, and the produced hitlists are compared with the performance and runtime of the CDPKit (Seidel, 2024) alignment algorithm. We further evaluate the pre-screening performance of our model and compare it against the PharmacoNet (Seo & Kim, 2023) model using the DEKOIS2.0 (Bauer et al., 2013) and LIT-PCBA (Tran-Nguyen et al., 2020) benchmark datasets.

**Benchmark datasets**    We perform experiments on the DUD-E benchmark dataset (Mysinger et al., 2012), which is commonly used to evaluate the performance of molecular docking and structure-based screening. The complete benchmark contains 102 protein targets, each accompanied by active and decoy ligands in the form of SMILES strings (Weininger, 1988) and the PDB template (Burley et al., 2017) of the ligand-receptor complex. We randomly select ten different protein targets for a proof-of-concept comparison with the CDPKit alignment algorithm. For our pre-screening experiment, we use the DEKOIS2.0 (Bauer et al., 2013) dataset, which contains 80 targets, each with 40 actives and 1,200 decoys, as well as the LIT-PCBA (Tran-Nguyen et al., 2020) dataset, consisting of 15 target sets with 7,761 confirmed actives and 382,674 inactive compounds. Database ligands are processed according to the data curation pipeline outlined in the Methodology section, with the exception that we sample up to 25 conformations per compound. Further information is provided in the Appendix A.6.

## 5.1    PHARMACOPHORIC FEATURE POINT PERCEPTION

We conduct a qualitative analysis through dimensionality reduction to gain a first intuition for the properties of the learned embedding space. The partial ordering of graph representations in the embedding space, based on the number of nodes per graph, is essential for encoding query-target relationships. This ordering property of the embedding space can be visualized using principal component analysis (PCA). Figure 4a displays the first two principal component axes of the learned representations, with the representations labeled according to the number of pharmacophoric feature points of the corresponding pharmacophores. This visualization demonstrates how the embedding vectors are systematically ordered relative to the number of nodes in each pharmacophore graph. Similarly, the Uniform Manifold Approximation and Projection (UMAP) algorithm (McInnes et al., 2020), a dimensionality reduction technique that preserves the local neighborhood structure of high-dimensional data, was employed. Figure 4b shows the UMAP representation of the embeddings, labeled by the number of feature points of a specific type. This visualization suggests that pharmacophores with a similar set of points are mapped proximally within the embedding space.

## 5.2    POSITIONAL PERCEPTION

We define a family of augmentations $\mathcal{T}_{r_D}$ to randomly delete nodes from a pharmacophore $P$ and displace the remaining nodes by a radius $r_D$. We sample augmentations $t_{r_D}(\cdot) \sim \mathcal{T}_{r_D}$ with increasing radius $r_D$ taken from a uniform grid of $m$ distances between 0 and 10 Å. For a given batch of pharmacophores $\{P_1, ..., P_n\}$, we generate the query-target pairs $\{(t_{r_D}(P_1), P_1), ..., (t_{r_D}(P_n), P_n)\}$. We then evaluate the decision function $g(\cdot, \cdot)$ (Equation 2) on the corresponding vector representations and calculate the mean of the decision function across all pairs against an increasing radius $r_D$, which is illustrated in Figure 4c. Without node displacement, the mean matching decision function is close to 1, indicating that the model recognizes pharmacophores with reduced node sets as valid queries. With a displacement of approximately 1.5 Å, the mean matching decision value drops to 50%, demonstrating the model's consideration of the chosen tolerance radius. Beyond a displacement of 1.5 Å, the decision function further decreases, approaching a plateau at approximately 6 Å. The results show that our model integrates 3D-positional information of pharmacophoric feature points into the learned representations.

## 5.3    ALIGNMENT PREDICTION AND PERFORMANCE AS A PRE-SCREENING TOOL

**Hitlist ranking**    Each benchmark set is comprised of a pharmacophore query $P_Q$ and a set of ligands $\mathcal{L} = \{L_1, ..., L_n\}$, where each ligand $L_i$ is associated with a set of pharmacophores

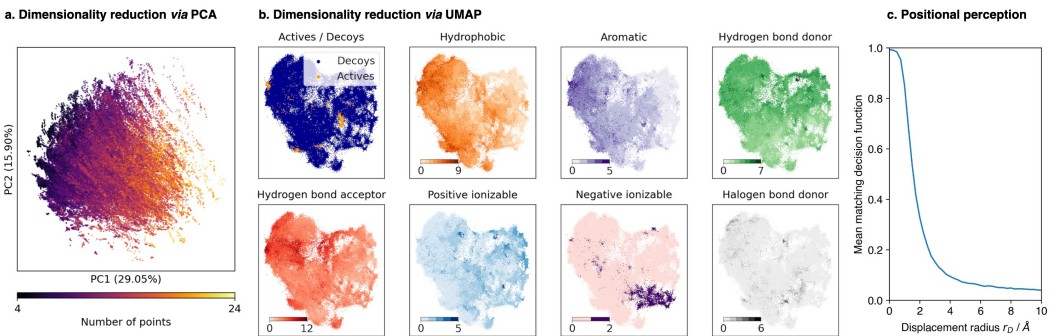

Figure 4: (a.) Dimensionality reduction of the ADA target's embedding space via PCA, with embeddings labeled by pharmacophoric feature point count. (b.) Dimensionality reduction via UMAP, with embeddings labeled by pharmacophoric feature point type. (c.) Experimental validation of the model's perception of 3D point positions, showing the mean matching decision function versus the displacement radius $r_D$ of the augmentation, with a decision threshold set to $t = 6500$.

$\{P_1, ..., P_{k_i}\}_i$ and a label $y_i$, which indicates whether the ligand is active or decoy. The task is to rank the database ligands *w.r.t.* the query, based on a scoring function $F : \mathcal{P} \times \mathcal{P} \to \mathbb{R}_+$. The ranking score $\psi_i$ of ligand $L_i$ is calculated through aggregation of the pharmacophore scores $\bigoplus(\{F(P_Q, P_1), ..., F(P_Q, P_{k_i})\}_i)$, where $\bigoplus$ is an aggregation operator. PharmacoMatch transforms the query $G(P_Q) \mapsto \mathbf{z}_Q$ and the set of pharmacophores $\{G(P_1), ..., G(P_{k_i})\}_i \mapsto \{\mathbf{z}_1, ..., \mathbf{z}_{k_i}\}_i$ *via* encoder model $f_\Theta : \mathcal{G} \to \mathbb{R}_+^D$ and evaluates the penalty function $E : \mathbb{R}_+^D \times \mathbb{R}_+^D \to \mathbb{R}_+$. A low penalty corresponds to a high ranking. The ranking score of database ligand $L_i$ is calculated as $\psi_i = \min(\{E(\mathbf{z}_Q, \mathbf{z}_1), ..., E(\mathbf{z}_Q, \mathbf{z}_{k_i})\}_i)$.

**Ground truth for alignment prediction** We evaluate the alignment prediction performance of PharmacoMatch by relative comparison with the alignment algorithm implemented in the open-source software CDPKit (Seidel, 2024), which utilizes clique-detection followed by Kabsch alignment (Kabsch, 1976). The ligand-receptor complex of the respective DUD-E target is used to generate a structure-based pharmacophore query with the CDPKit. The CDPKit alignment algorithm only returns exact matches, meaning that queries with an excessive number of points may yield no results. Consequently, pharmacophore modeling often requires user interaction to reduce the number of points in the query. For our comparison, we refined the initial query pharmacophores to a subset of 5–7 pharmacophoric feature points, which is a common range in pharmacophore modeling. These points were selected to ensure that the query yields meaningful enrichment for the CDPKit algorithm. Only after this refinement did we proceed to compare PharmacoMatch against this ground truth. The alignment of a query $P_Q$ and a target $P_T$ is evaluated with an *alignment score* $S : \mathcal{P} \times \mathcal{P} \to \mathbb{R}_+$, which takes into account the number of matched features and their geometric fit (further details are provided in the Appendix A.6). The ligand ranking score is calculated as $\psi_i = \max(\{S(P_Q, P_1), ..., S(P_Q, P_{k_i})\}_i)$, the highest alignment score represents the score for the database ligand. Analogous to equation (2), we can also define a matching decision function $\phi$ based on the alignment score, where $t = |P_Q|$:

$$\phi(P_Q, P_T) = \begin{cases} 1 & \text{iff } S(P_Q, P_T) \geq t \\ 0 & \text{otherwise} \end{cases} \tag{3}$$

**Evaluation metrics** Both algorithms rank database ligands to produce a *hitlist*. We assess the performance of PharmacoMatch on the benchmark using two approaches. First, we demonstrate that the PharmacoMatch penalty $E(\cdot, \cdot)$ correlates with the matching decision function $\phi(\cdot, \cdot)$ of the alignment algorithm. We evaluate both functions against all pharmacophores in a dataset *w.r.t.* query $P_Q$. The outputs are compared by generating the corresponding receiver operating characteristic (ROC) curves, and the performance is quantified using the area under the ROC curve (AUROC) metric. We argue that this relative performance is the key metric for evaluating the alignment prediction performance of our model. Second, we compare the absolute screening performance of our

Table 1: Method comparison and screening performance of the PharmacoMatch algorithm and the CDPKit alignment algorithm on ten different DUD-E protein targets (see Appendix A.6 for details). BEDROC values are calculated with $\alpha = 20$, as recommended by Truchon & Bayly (2007), AUROC and BEDROC are reported in percent. Confidence intervals are calculated using bootstrapping (Efron, 1979), with standard deviations reported based on 100 resampled datasets.

| Protein target | Relative performance | Absolute screening performance PharmacoMatch | | | | | Absolute screening performance CDPKit | | | | |
|---|---|---|---|---|---|---|---|---|---|---|---|
| | AUROC | AUROC | BEDROC | $EF_{1\%}$ | $EF_{5\%}$ | $EF_{10\%}$ | AUROC | BEDROC | $EF_{1\%}$ | $EF_{5\%}$ | $EF_{10\%}$ |
| ACES | $96.0 \pm 0.2$ | $58 \pm 2$ | $18 \pm 1$ | $8.4 \pm 1.4$ | $3.5 \pm 0.3$ | $2.2 \pm 0.2$ | $55 \pm 1$ | $16 \pm 2$ | $5.5 \pm 1.3$ | $3.0 \pm 0.3$ | $2.1 \pm 0.2$ |
| ADA | $98.3 \pm 0.2$ | $83 \pm 3$ | $44 \pm 4$ | $16.7 \pm 4.1$ | $9.5 \pm 1.0$ | $5.7 \pm 0.4$ | $94 \pm 1$ | $82 \pm 3$ | $53.6 \pm 4.3$ | $15.9 \pm 0.9$ | $8.4 \pm 0.4$ |
| ANDR | $98.8 \pm 0.1$ | $76 \pm 1$ | $33 \pm 2$ | $15.8 \pm 1.9$ | $6.0 \pm 0.5$ | $4.3 \pm 0.3$ | $71 \pm 2$ | $26 \pm 2$ | $12.6 \pm 2.1$ | $4.4 \pm 0.5$ | $3.7 \pm 0.3$ |
| EGFR | $91.1 \pm 0.5$ | $63 \pm 1$ | $11 \pm 1$ | $3.1 \pm 0.7$ | $2.0 \pm 0.3$ | $1.6 \pm 0.2$ | $76 \pm 1$ | $26 \pm 2$ | $12.2 \pm 1.6$ | $4.6 \pm 0.3$ | $3.7 \pm 0.2$ |
| FA10 | $84.3 \pm 0.1$ | $47 \pm 1$ | $1 \pm 1$ | $0.2 \pm 0.2$ | $0.1 \pm 0.1$ | $0.2 \pm 0.1$ | $55 \pm 1$ | $6 \pm 1$ | $0.0 \pm 0.0$ | $0.7 \pm 0.2$ | $1.2 \pm 0.1$ |
| KIT | $84.5 \pm 0.1$ | $56 \pm 2$ | $4 \pm 1$ | $0.0 \pm 0.0$ | $0.4 \pm 0.2$ | $0.7 \pm 0.2$ | $63 \pm 2$ | $9 \pm 2$ | $1.1 \pm 0.8$ | $1.2 \pm 0.4$ | $1.8 \pm 0.3$ |
| PLK1 | $79.5 \pm 0.5$ | $62 \pm 3$ | $9 \pm 2$ | $1.5 \pm 1.3$ | $0.7 \pm 0.3$ | $1.8 \pm 0.3$ | $75 \pm 3$ | $39 \pm 3$ | $5.7 \pm 2.3$ | $10.2 \pm 0.9$ | $5.5 \pm 0.5$ |
| SRC | $96.6 \pm 0.2$ | $79 \pm 1$ | $27 \pm 1$ | $6.0 \pm 1.0$ | $5.3 \pm 0.4$ | $4.6 \pm 0.2$ | $80 \pm 1$ | $28 \pm 1$ | $11.1 \pm 1.2$ | $5.3 \pm 0.4$ | $4.3 \pm 0.2$ |
| THRB | $89.9 \pm 0.5$ | $70 \pm 1$ | $22 \pm 1$ | $5.9 \pm 1.0$ | $4.8 \pm 0.4$ | $3.3 \pm 0.2$ | $79 \pm 1$ | $35 \pm 2$ | $11.8 \pm 1.5$ | $7.2 \pm 0.4$ | $4.5 \pm 0.2$ |
| UROK | $83.8 \pm 0.2$ | $60 \pm 2$ | $4 \pm 1$ | $0.6 \pm 0.7$ | $0.5 \pm 0.2$ | $0.4 \pm 0.2$ | $91 \pm 1$ | $55 \pm 3$ | $24.5 \pm 2.8$ | $10.4 \pm 0.9$ | $8.2 \pm 0.4$ |

model and the alignment algorithm using the ligand ranking score $\psi$. The primary objective of virtual screening is to find active compounds among decoys. The AUROC metric is used to evaluate the overall classification performance *w.r.t.* activity label $y_i$. A drawback of this metric is that it does not reflect the early enrichment of active compounds in the hitlist, which is of significant interest in virtual screening. Early enrichment is assessed using the enrichment factor ($EF_{\alpha\%}$) and the Boltzmann-enhanced discrimination of ROC (BEDROC) metric (Truchon & Bayly, 2007), which assigns higher weights to better-ranked samples (definitions in Appendix A.6). Note that these performance metrics are entirely dependent on the chosen query. Therefore, our primary performance metric in this experiment is the relative performance AUROC. For completeness, we also report absolute virtual screening metrics. We conduct this proof-of-concept comparison using a randomly selected subset of 10 DUD-E targets, serving as a focused case study.

**Alignment prediction performance**  Our results, comparing PharmacoMatch with the CDPKit alignment algorithm across the selected targets, are summarized in Table 1 (ROC plots are provided in the Appendix A.6). We observe a robust correlation between the hitlists generated by the two algorithms, demonstrating the effectiveness of our approach. This correlation varies by target, reflecting the sensitivity of virtual screening to the chosen query. Although the alignment algorithm achieves generally higher AUROC scores and early enrichment, our method consistently produces hitlists with competitive performance across several targets. In terms of runtime, PharmacoMatch significantly outperforms the alignment algorithm. We compare the time required for alignment, embedding, and vector matching per pharmacophore. Alignment is performed in parallel on an AMD EPYC 7713 64-Core Processor with 128 threads, while pharmacophore embedding and matching are run on an NVIDIA GeForce RTX 3090, with both devices having comparable purchase prices and release dates. Creating vector embeddings from pharmacophore graphs takes $67 \pm 7$ $\mu$s per pharmacophore, approximately as long as aligning a query to a target with $66 \pm 6$ $\mu$s. However, the embedding process only needs to be performed once. Subsequently, the preprocessed vector data can be used for vector matching, which takes $0.3 \pm 0.1$ $\mu$s, being approximately two orders of magnitude faster than the alignment. Additionally, vector comparison is independent of the query size, an advantage not shared by the alignment algorithm. Although executed on different hardware, this comparison highlights the speed-gain of our algorithm.

**Applicability as a pre-screening tool**  We demonstrated how PharmacoMatch predicts pharmacophore matching and compared its performance with the CDPKit alignment algorithm. A key aspect of this comparison is the role of user interaction in query design, since the alignment algorithm only identifies a match when all pharmacophoric feature points align, and while it is possible to define feature points as optional, this flexibility significantly increases runtime due to the combinatorial explosion of possible query patterns. Consequently, handling larger queries with ten or

Table 2: Comparison of PharmacoMatch with PharmacoNet (Seo & Kim, 2023) as a pre-screening tool. The reported performance metrics were averaged over all targets of the DEKOIS2.0 and LIT-PCBA datasets. AUROC and BEDROC are given in percent, BEDROC values are calculated with $\alpha = 80.5$, as reported in Seo & Kim (2023). PharmacoNets runtime per ligand depends on the number of atoms, here we report their measurement for ligands with 70 heavy atoms. The runtime for PharmacoMatch was calculated as average over all ligands in the benchmark.

| | DEKOIS2.0 | | | | | LIT-PCBA | | | | | Runtime |
|---|---|---|---|---|---|---|---|---|---|---|---|
| | AUROC | BEDROC | $EF_{0.5\%}$ | $EF_{1\%}$ | $EF_{5\%}$ | AUROC | BEDROC | $EF_{0.5\%}$ | $EF_{1\%}$ | $EF_{5\%}$ | per ligand (s) |
| PharmacoNet | **62.5** | 12.3 | 4.4 | 4.2 | 2.9 | - | - | - | 3.1 | - | $5.2 \cdot 10^{-3}$ |
| PharmacoMatch (ours) | 60.9 | **15.1** | **5.5** | **4.9** | **3.2** | 57.4 | 5.0 | 6.0 | **3.5** | 2.2 | $\mathbf{3.3 \cdot 10^{-6}}$ |

more points becomes computationally prohibitive. PharmacoMatch operates independently of the query's size. This characteristic makes PharmacoMatch particularly suitable as a pre-screening tool. By using structure-based queries without refinement step, it can filter extensive datasets effectively, reducing the computational burden before engaging more resource-intensive methods or requiring user interaction. To evaluate this use case, we compare our method with the recent pre-screening tool PharmacoNet (Seo & Kim, 2023), which employs image segmentation to generate pharmacophore queries and a parameterized analytical function for hitlist creation. To replicate an automated pre-screening scenario, we exclude user interaction from our workflow. Specifically, we generate an interaction pharmacophore using CDPKit and use it directly as a query for PharmacoMatch. We assess the pre-screening performance of PharmacoMatch and PharmacoNet on the DEKOIS2.0 (Bauer et al., 2013) and LIT-PCBA (Tran-Nguyen et al., 2020) benchmark datasets. Performance is evaluated using the average AUROC, BEDROC, and EF metrics across all targets (see Appendix A.6 for more details). The results of this comparison are summarized in Table 2. Our evaluation demonstrates that PharmacoMatch outperforms PharmacoNet with respect to early enrichment. Importantly, PharmacoMatch's runtime is three orders of magnitude faster. Given that the primary goal of pre-screening is rapid and cost-effective filtering before using more computationally expensive methods, we argue that this substantial improvement in runtime makes PharmacoMatch an effective pre-screening tool.

**Practical considerations & limitations** There are two options for integrating our model into a virtual screening pipeline. First, the PharmacoMatch model can be used in place of the alignment algorithm to generate a hitlist of potential active compounds, which is suitable for quickly producing a compound list for experimental testing. Second, our method can serve as an efficient pre-screening tool for very large databases, reducing the number of molecules from billions to millions, after which the slower alignment algorithm can be applied to this filtered subset. Note that alignment will still be necessary if visual inspection of aligned pharmacophores and corresponding ligands is desired. An important limitation is that the embedding process is not lossless and might lead to reduced 3D geometric precision, when compared to an alignment algorithm. The employed E(3)-invariant encoder cannot distinguish a pharmacophore from its mirror image, potentially increasing the false positive rate. Addressing these limitations will be part of future work.

## 6    CONCLUSION

We have presented PharmacoMatch, a contrastive learning framework that creates meaningful pharmacophore representations for virtual screening. The proposed method tackles the matching of 3D pharmacophores through vector comparison in an order embedding space, thereby offering a valuable method for significant speed-up of virtual screening campaigns. PharmacoMatch is the first machine-learning based solution that approaches pharmacophore virtual screening *via* an approximate neural subgraph matching algorithm. We are confident that our method will help to improve on existing virtual screening workflows and contribute to the assistance of medicinal chemist in the complex task of drug discovery.

REPRODUCIBILITY STATEMENT

The source code of this project can be found under the following link: `https://github.com/molinfo-vienna/PharmacoMatch`. Please follow the instructions in the repository for reproduction of our results. Training and test data can be downloaded here: `https://doi.org/10.6084/m9.figshare.27061081`.

ACKNOWLEDGEMENT

We thank Roxane Jacob, Nils Morten Kriege, and Christian Permann from the University of Vienna, Klaus-Jürgen Schleifer from BASF SE, and Andreas Bergner from Boehringer-Ingelheim RCV GmbH & Co KG for fruitful discussions and proof-reading of the manuscript. We also thank the anonymous reviewers for their suggestions and comments. Financial support received for the Christian Doppler Laboratory for Molecular Informatics in the Biosciences by the Austrian Federal Ministry of Labour and Economy, the National Foundation for Research, Technology and Development, the Christian Doppler Research Association, Boehringer-Ingelheim RCV GmbH & Co KG and BASF SE is gratefully acknowledged.

AUTHOR CONTRIBUTIONS

Conceptualization: DR, OW. Methodology: DR, OW. Data curation & analysis: DR. Code implementation & model training: DR. CDPKit software & support: TS. Investigation: DR. Writing (original draft): DR. Writing (review and editing): DR, OW, TS, TL. Rebuttal: DR. Funding acquisition: TL. Resources: TL. Supervision: OW, TL. All authors have given approval to the final version of the manuscript.

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

## A  APPENDIX

### A.1  DATASET CURATION & STATISTICS

Unlabeled training data was downloaded from the ChEMBL database to represent small molecules with drug-like properties. At the time of data download, the ChEMBL database contained 2,399,743 unique compounds. We constrained the compound category to "small molecules" and enforced adherence to the Lipinsky rule of five (Lipinski et al., 1997), specifically setting violations to "0," resulting in a refined set of 1,348,115 compounds available for download. The molecules were acquired in the form of Simplified Molecular Input Line Entry System (SMILES) (Weininger, 1988) strings. Subsequent to data retrieval, we conducted preprocessing using the database cleaning functionalities of the Chemical Data Processing Toolkit (CDPKit) (Seidel, 2024). This process involved the removal of solvents and counter ions, adjustment of protonation states to a physiological pH value, and elimination of duplicate structures, where compounds differing only in their stereo configuration were regarded as duplicates. To prevent data leakage, we carefully removed all structures from the training data that would occur in one of the test sets we used for our benchmark experiments. The final set was comprised of 1,221,098 compounds. For each compound within the dataset, a 3D conformation was generated using the CONFORGE (Seidel et al., 2023) conformer generator from the CDPKit, which was successful for 1,220,104 compounds. To enhance batch diversity, we generated only one conformation per compound for contrastive training. Subsequently, 3D pharmacophores were computed for each conformation, with removal of pharmacophores containing less than four pharmacophoric feature points. The ultimate dataset comprised 1,217,361 distinct pharmacophores. Figure 5 shows the frequency of pharmacophores with a specific pharmacophoric feature point count in the training data. On average, a pharmacophore consists of 12.4 pharmacophoric feature points, with the largest pharmacophore in the dataset containing 32 points. Hydrophobic feature points and hydrogen bond acceptors are the most prominent, while hydrogen bond donors and aromatics occur less frequently. Ionizable feature points and halogen bond donors are comparatively rare.

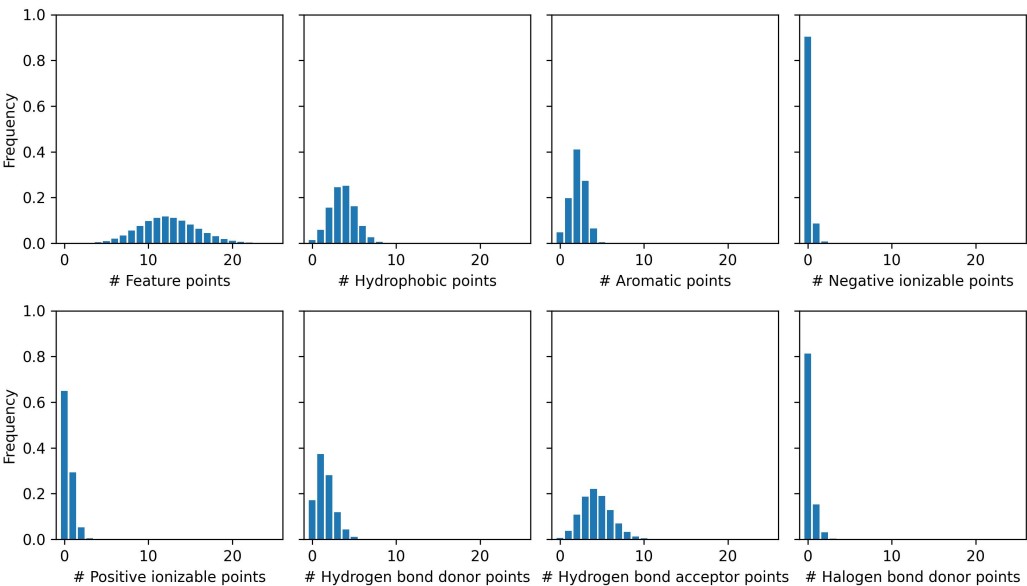

Figure 5: Pharmacophoric feature point statistics of the training data. The respective histograms display the total number of pharmacophoric feature points and the number of points of specific types per pharmacophore in the training data. The complete training dataset contains 1,217,361 distinct pharmacophores.

## A.2 Augmentation module

The augmentation module receives the initial pharmacophore $\mathbf{x_0} = [\mathbf{h_0}, \mathbf{r_0}]$, with the initial OHE feature matrix $\mathbf{h_0}$ and the Cartesian coordinates $\mathbf{r_0}$. Edge attributes of the complete graph were calculated from the pair-wise distances between nodes after modifying the input according to the augmentation strategy, which combines random node deletion and random node displacement. The module outputs the modified tuple $\mathbf{x} = [\mathbf{h}, \mathbf{e}]$ with the feature matrix $\mathbf{h}$ and the edge attributes $\mathbf{e}$.

**Node deletion**   Random node deletion involved removing at least one node, with the upper bound determined by the cardinality of the set of nodes $V_i$ of graph $G_i$. To ensure the output graph retained at least three nodes, the maximum number of deletable nodes was $|V_i| - 3$. The number of nodes to delete was drawn uniformly at random.

**Node displacement**   There are two modes for the displacement of pharmacophoric feature points, displacement within the tolerance sphere, and displacement onto the surface of the tolerance sphere. For simplicity, we assumed the same tolerance sphere radius $r_T$ across different pharmacophoric feature types. For displacement within the tolerance sphere, we calculated the coordinate displacement $(\Delta x, \Delta y, \Delta z)$ from spherical coordinates $\phi \sim \mathcal{U}(0, 2\pi)$ and $\cos\theta \sim \mathcal{U}(-1, 1)$, which were drawn at random from a uniform distribution:

$$\Delta x = \Delta r \sin\theta \cos\phi, \ \Delta y = \Delta r \sin\theta \sin\phi, \ \Delta z = \Delta r \cos\theta \tag{4}$$

where $\Delta r = r_T \sqrt[3]{u}$ and $u \sim \mathcal{U}(0, 1)$. Displacement of the nodes onto the tolerance sphere surface was achieved by calculating the mean of the positions of the pharmacophoric feature points, $\mu = \frac{1}{|P|} \sum_{i=1}^{|P|} \mathbf{r}_i$, and displacing each point $p_i$ by $r_T$ in the direction $\mathbf{r}_i - \mu$. The displacement away from the center ensures that displacement directions do not cancel each other.

## A.3 Message passing neural network

Convolution on irregular domains like graphs is formulated as message passing, which can generally be described as:

$$\mathbf{h}_i^{(k)} = \gamma^{(k)}(\mathbf{h}_i^{(k-1)}, \bigoplus_{j \in \mathcal{N}(i)} \phi^{(k)}(\mathbf{h}_i^{(k-1)}, \mathbf{h}_j^{(k-1)}, \mathbf{e}_{ij})) \tag{5}$$

where $\mathbf{h}_i^{(k)} \in \mathbb{R}^{F'}$ denotes the node features of node $i$ at layer $k$, $\mathbf{h}_i^{(k-1)} \in \mathbb{R}^F$ denotes the node features of node $i$ at layer $k - 1$, $\mathbf{e}_{ij} \in \mathbb{R}^D$ the edge features of the edge from node $i$ to node $j$, $\gamma^{(k)}$ and $\phi^{(k)}$ are parameterized, differentiable functions, and $\bigoplus$ is an aggregation operator like, *e. g.*, the summation operator (Fey & Lenssen, 2019). In our encoder architecture, we employed the following edge-conditioned convolution operator, which was proposed both by Gilmer et al. (2017) and Simonovsky & Komodakis (2017):

$$\mathbf{h}_i^{(k)} = \Theta\mathbf{h}_i^{(k-1)} + \sum_{j \in \mathcal{N}(i)} \mathbf{h}_j^{(k-1)} \cdot \psi_\Theta(\mathbf{e}_{ij}) \tag{6}$$

where $\Theta \in \mathbb{R}^{F \times F'}$ denotes learnable weights and $\psi_\Theta(\cdot) : \mathbb{R}^D \to \mathbb{R}^{F \times F'}$ denotes a neural network, in our case an MLP with one hidden layer. These transformations map node features $\mathbf{h}$ into a latent representation that combines pharmacophoric feature types with distance encodings.

## A.4 Encoder implementation

The encoder was implemented as a GNN $f_\Theta : \mathcal{G} \to \mathbb{R}_+^D$ that maps a given graph $G$ to the abstract representation vector $\mathbf{z} \in \mathbb{R}_+^D$. The architecture is comprised of an initial embedding block, three subsequent convolution blocks, followed by a pooling layer, and a projection block.

**Embedding block**   The embedding block receives the pharmacophore graph $G_i$ as the tuple $\mathbf{x}_i = [\mathbf{h}_i, \mathbf{e}_i]$, with the OHE feature matrix $\mathbf{h}_i$ and the edge attributes $\mathbf{e}_i$. Initial node feature embeddings are created from the OHE features with a fully-connected (FC) dense layer with learnable weights $\mathbf{W}$ and bias $\mathbf{b}$:

$$\mathbf{h}_i \leftarrow \mathbf{W}\mathbf{h}_i + \mathbf{b} \tag{7}$$

**Convolution block**   The convolution block consists of a graph convolution layer, which is implemented as edge-conditioned convolution operator (NNConv), the update rule is described in Section A.3. The network further consists of batch normalization layers (BN), GELU activation functions, and dropout layers. The hidden representation $\mathbf{h}_i^l$ of graph $G_i$ is updated at block $l$ as follows:

$$[\mathbf{h}_i^l, \mathbf{e}_i] \rightarrow \{\text{NNConv} \rightarrow \text{BN} \rightarrow \text{GELU} \rightarrow \text{concat}(\mathbf{h}_i^{l'}, \mathbf{h}_i^l) \rightarrow \text{dropout}\} \rightarrow \mathbf{h}_i^{l+1} \tag{8}$$

where $\mathbf{h}_i^{l'}$ represents the latent representation after activation. Updating the feature matrix $l$ times yields the final node representations of the pharmacophoric feature points.

**Pooling layer**   We employed additive pooling for graph-level read-out $\mathbf{r}_i$, which aggregates the set of $|V|$ node representations $\{\mathbf{h}_1, ..., \mathbf{h}_{|V|}\}_i$ of a Graph $G_i$ by element-wise summation:

$$\mathbf{q}_i = \sum_{k=1}^{|V|} \mathbf{h}_k \tag{9}$$

**Projection block**   The projection block maps the graph-level read-out to the positive real number space and is implemented as a multi-layer perceptron $\text{MLP} : \mathbb{R}^d \rightarrow \mathbb{R}_+^D$, where $d$ is the dimension of the vector representation before and $D$ the dimension after the projection. The block consists of $k$ sequential layers of FC layers, BN, ReLU activation, and dropout:

$$\mathbf{q}_i^k \rightarrow \{\text{FC} \rightarrow \text{BN} \rightarrow \text{ReLU} \rightarrow \text{Dropout}\} \rightarrow \mathbf{q}_i^{k+1} \tag{10}$$

The final layer is a FC layer without bias and with positive weights, only:

$$\mathbf{z}_i \leftarrow \text{abs}(\mathbf{W})\mathbf{q}_i \tag{11}$$

Matrix multiplication of the positive learnable weights $\mathbf{W}$ and the output of the last ReLU activation function produces the final representation $\mathbf{z}_i \in \mathbb{R}_+^D$.

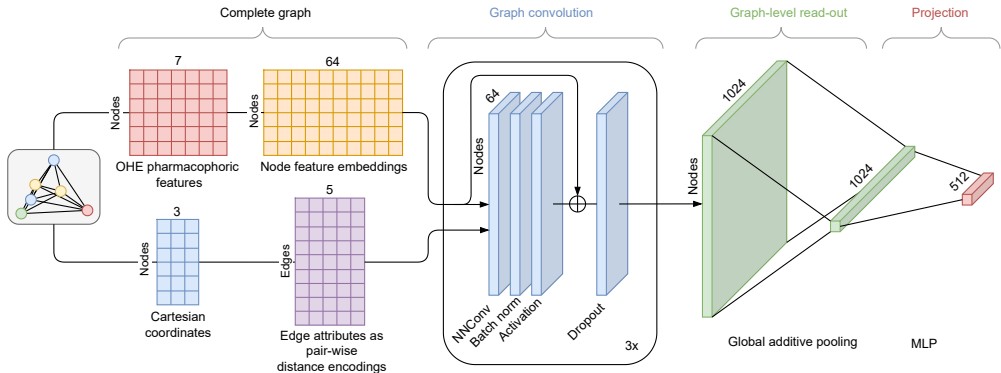

Figure 6: Architecture of the GNN encoder model.

A.5   Model implementation and training

**Implementation dependencies**   The GNN was implemented in Python 3.10 with PyTorch (v2.0.1) and the PyTorch Geometric library (v2.3.1) (Fey & Lenssen, 2019). Both, model and dataset, were implemented within the PyTorch Lightning (Falcon & The PyTorch Lightning team, 2019) framework (v2.1.0). Model training was monitored with Tensorboard (v2.13.0). CDPKit (v1.1.1) was employed for chemical data processing. Software was installed and executed on a Rocky Linux (v9.4) system with x86-64 architecture.

**Model training**   Training was performed on a single NVIDIA GeForce 3090 RTX graphics unit with 24 GB GDDR6X. Training runs were performed for a maximum of 500 epochs with a batch size of 256 pharmacophore graphs. Curriculum learning was applied by gradual enrichment of the dataset with increasingly larger pharmacophore graphs. At training start, only pharmacophore graphs with 4 nodes were considered. After 10 subsequent epochs without considerable minimization of the loss function, pharmacophore graphs with one additional node were added to the training data. This approach allows the model to start with very simple examples, gradually increasing the difficulty of the matching task. The loss function was minimized with the Adam (Kingma, 2014) optimizer, we further applied gradient clipping. A training run on the full dataset took approximately 48 hours with the above hardware specifications.

**Hyperparameter optimization & model selection**   Hyperparameters were optimized through random parameter selection, the tested ranges are summarized in Table 4. Unlabeled data was split into training and validation data with a 98:2 ratio. Training runs were compared using the AUROC value on the validation data. This was calculated by treating the positive and negative pairs as binary labels, and the predictions were based on their respective order embedding penalty, which was calculated with function $E(\cdot, \cdot)$ in Equation (1). Hyperparameter optimization was performed on a reduced dataset with 100,000 graphs, which took approximately 5 hours per run. The best performing models were retrained on the full dataset. The hyperparameters of the final encoder model are summarized in Table 3. After model selection, the final model performance was tested on virtual screening datasets.

Table 3: Hyperparameters of the best performing encoder model

| Hyperparameter | |
| --- | --- |
| batch size | 256 |
| dropout convolution block | 0.2 |
| dropout projection block | 0.2 |
| max. epochs | 500 |
| hidden dimension convolution block | 64 |
| hidden dimension projection block | 1024 |
| output dimension convolution block | 1024 |
| output dimension projection block | 512 |
| learning rate optimizer | 0.001 |
| margin for negative pairs | 100.0 |
| number of convolution blocks | 3 |
| depth of the projector MLP | 3 |
| edge attributes dimension | 5 |
| sampling sphere radius positive pairs | 1.5 |
| sampling surface radius negative pairs | 1.5 |

**Ablation studies**   To evaluate the importance of various model parameters, we conduct a series of ablation studies using the best-performing model. In these experiments, we systematically alter one parameter at a time and assess its impact on classification performance using the validation hold-out set. Our findings reveal several key insights. The embedding dimension of the learned representations can be reduced to 128 without loss in performance. The encoder requires at least 32 dimensions to remain effective. Skip-connections are critical for model performance, with DenseNet-style connections slightly outperforming ResNet-style (He et al., 2015) alternatives. Interestingly, the choice

Table 4: Tested hyperparameter ranges for model training.

| Hyperparameter | Parameter range |
|---|---|
| dropout | [0.2, 0.3, 0.4, 0.5] |
| margin for negative pairs | [0.1, 0.5, 1, 2, 5, 10, 100, 1000] |
| output dimension projection block | [64, 128, 256, 512, 1024] |
| displacement sphere radius $r_T$ of positive pairs | [0.25, 0.5, 1.0, 1.5] |

Table 5: Tested hyperparameter ranges for ablation studies.

| Parameter | Parameter range | Validation AUROC |
|---|---|---|
| Tolerance radius | [0.0, 0.5, 1.0, 1.5, 2.0] | [0.91, 0.93, 0.94, 0.94, 0.93] |
| Encoder dimension | [8, 16, 32, 64, 96] | [0.92, 0.93, 0.94, 0.94, 0.94] |
| Embedding dimension | [32, 64, 128, 256, 512, 1024] | [0.91, 0.93, 0.94, 0.94, 0.94, 0.93] |
| Skip-connection | [dense, res, none] | [0.94, 0.93, 0.74] |
| GNN Layer | [NNConv, GINE, CFConv, GAT] | [0.94, 0.94, 0.94, 0.93] |
| Projector layers | [1, 2, 3] | [0.94, 0.94, 0.94] |
| Convolution layers | [1, 2, 3] | [0.94, 0.94, 0.94] |
| Margin | [0.01, 0.1, 1, 2, 5, 10, 100, 1000] | [0.88, 0.90, 0.92, 0.92, 0.93, 0.93, 0.94, 0.94] |

of message-passing layer, whether NNConv, graph isomorphism operator (GINE) (Hu et al., 2020), graph attention operator (GAT) (Brody et al., 2022), or continuous-filter convolutional layers (CF-Conv) (Schütt et al., 2018), has minimal impact on performance. The depth of the projector and encoder also does not significantly affect results. In contrast, the margin value plays a significant role in model performance. While larger values enhance performance, excessively high margins can lead to training instability. A margin of 100 provides an effective balance between these factors. The displacement radius for augmentations in creating positive pairs is most effective at 1.5 Å, while removing node displacement degrades model performance.

## A.6 VIRTUAL SCREENING

**DUD-E dataset details** General information about the DUD-E targets is summarized in Table 6. For each target we downloaded the receptor structure from the PDB and created the corresponding interaction pharmacophore with the CDPKit. Vector features were converted into undirected pharmacophoric feature points. The resulting pharmacophore queries (Figure 7) were used in our virtual screening experiments.

Table 6: DUD-E targets that were selected for bechmarking experiments in this study.

| Target | PDB code | Ligand ID | Active Ligands | Active Conformations | Decoy Ligands | Decoy Conformations | Query Points |
|---|---|---|---|---|---|---|---|
| ACES | 1e66 | HUX | 451 | 10048 | 26198 | 567122 | 6 |
| ADA | 2e1w | FR6 | 90 | 2166 | 5448 | 125035 | 7 |
| ANDR | 2am9 | TES | 269 | 3039 | 14333 | 211968 | 6 |
| EGFR | 2rgp | HYZ | 541 | 12468 | 35001 | 755017 | 7 |
| FA10 | 3kl6 | 443 | 537 | 13343 | 28149 | 638831 | 5 |
| KIT | 3g0e | B49 | 166 | 3703 | 10438 | 224364 | 5 |
| PLK1 | 2owb | 626 | 107 | 2531 | 6794 | 152999 | 6 |
| SRC | 3el8 | PD5 | 523 | 11868 | 34407 | 737864 | 6 |
| THRB | 1ype | UIP | 461 | 11494 | 26894 | 626722 | 7 |
| UROK | 1sqt | UI3 | 162 | 3450 | 9837 | 199204 | 6 |

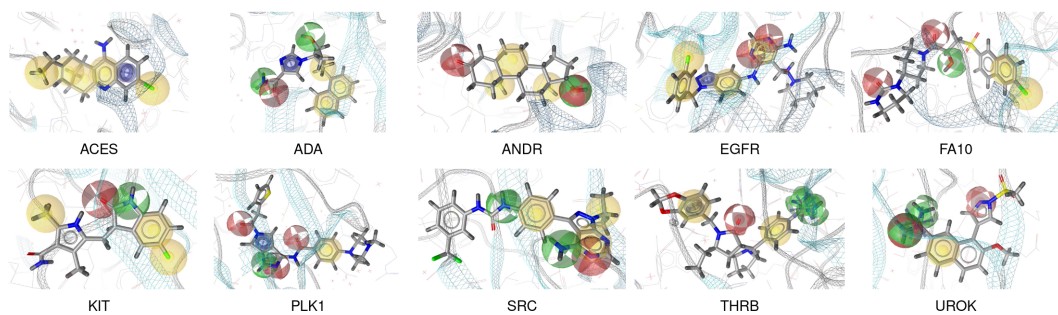

Figure 7: Structure-based pharmacophore queries of ten targets of the DUD-E benchmark dataset.

**DEKOIS2.0 dataset details**   General information about the DEKOIS2.0 targets (Bauer et al., 2013). Each target is associated with a PDB four-letter code and a corresponding ligand ID. For each target, we downloaded the receptor structure and its respective ligand from the PDB and generated the interaction pharmacophore using the CDPKit. These queries were used without further refinement in our pre-screening experiments. For targets containing the small molecule ligand in multiple binding pockets, we randomly selected one pocket for pharmacophore generation. The SIRT2 target was excluded from our evaluation because its structure does not contain a small molecule ligand. Actives and decoys were processed analogous to the ligands in the DUD-E benchmark.

**LIT-PCBA dataset details**   For each target, the downloaded files include a SMILES file containing the active and inactive ligands, along with receptor structures provided as PDB files. To ensure a fair comparison with PharmacoNet, we created pharmacophore queries using the same receptor PDB files as in their study. These PDB files were selected based on a methodology described by (Shen et al., 2023) and are listed in Table S4 of the respective supporting information. Interaction pharmacophores were generated using CDPKit and used without further refinement in our pre-screening experiments. Active and inactive ligands were processed following the same protocol as for the DUD-E benchmark.

**CDPKit alignment scoring function**   The CDPKit implements alignment as a clique-detection algorithm and computes a rigid-body transformation *via* Kabsch's algorithm to align the pharmacophore query $P_Q$ to the pharmacophore target $P_T$. The goodness of fit is evaluated with a geometric scoring function $S : \mathcal{P} \times \mathcal{P} \to \mathbb{R}_+$:

$$S(P_Q, P_T) = S_{MFP}(P_Q, P_T) + S_{Geom}(P_Q, P_T) \tag{12}$$

where $S_{MFP} : \mathcal{P} \times \mathcal{P} \to \mathbb{N}$ counts the number of matched feature pairs and $S_{Geom} : \mathcal{P} \times \mathcal{P} \to [0, 1)$ evaluates their geometric fit.

**Runtime measurement**   We measured alignment runtimes using the psdscreen tool from the CDP-Kit with 128 threads on an AMD EPYC 7713 64-Core Processor, while embedding and matching runtimes with PharmacoMatch were recorded using an NVIDIA GeForce RTX 3090 GPU with 24 GB GDDR6X. We additionally used 8 CPU worker nodes for the embedding step. Runtime per pharmacophore was estimated by dividing the total runtime by the number of pharmacophores in each dataset, with the final estimate taken as the mean of ten runs. The results report the mean and standard deviation of these estimates across all ten datasets.

**Performance metrics**   The enrichment factor (EF) is formally defined as:

$$EF_\alpha = \frac{n_\alpha}{n \cdot \alpha} \tag{13}$$

with $n_\alpha$ the number of true active compunds in the top $\alpha\%$ of the hitlist ranking and the total number of active compounds $n$.

The Boltzmann-Enhanced Discrimination of Receiver Operating Characteristic (BEDROC) metric (Truchon & Bayly, 2007) is similar to the AUROC metric but assigns greater weight to samples ranked higher in the hitlist. The degree of this reweighting is controlled by the early recognition parameter $\alpha$. The formal definition is as follows:

$$BEDROC_\alpha = \frac{\sum_{i=1}^{n} e^{-\alpha r_i/N}}{R_a\left(\frac{1-e^{-\alpha}}{e^{\alpha/N}-1}\right)} \cdot \frac{R_a \sinh\left(\alpha/2\right)}{\cosh\left(\alpha/2\right) - \cosh\left(\alpha/2 - \alpha R_a\right)} + \frac{1}{1 - e^{\alpha(1-R_a)}} \tag{14}$$

where $n$ is the number of active compounds in the dataset, $N$ is the total number of compounds, $R_a = n/N$ is the ratio of active compounds in the dataset, and $r_i = \frac{\text{rank}_i - 1}{N-1}$ is the normalized ranking of the $i$-th active compound.

The area under receiver operating characteristic (AUROC) performance metrics of our virtual screening experiments on the selected DUD-E targets are derived from the ROC curves presented in Figures 8 and Figure 9. The performance metrics for the pre-screening experiments are calculated as follows: for each target, we calculated the AUROC, BEDROC, and enrichment factors (EF) for the top $0.5\%$, $1\%$, and $5\%$ of the hit list, reporting the average values for these metrics. Metrics for PharmacoNet were reported as presented in their original manuscript and were not recalculated.

### A.7 EMBEDDING SPACE VISUALIZATION

**UMAP visualization**   UMAP embeddings for visualization plots were calculated with the *UMAP* Python library. The 'metric' parameter was set to Manhattan distance, all other parameters are the default settings of the implementation. We tested a range of hyperparameters to ensure that the visualization results are not sensitive to parameter selection.

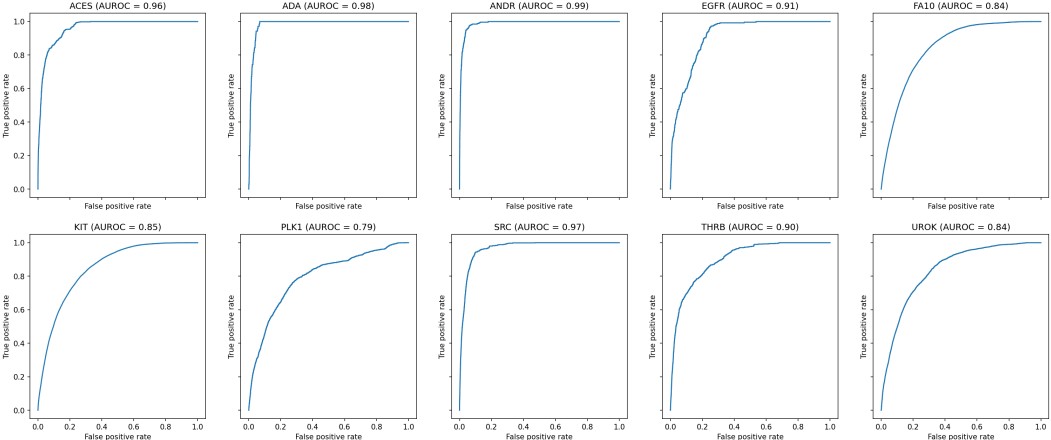

Figure 8: Performance comparison of PharmacoMatch and the alignment algorithm. The ROC-curves display the agreement of the hitlist ranking of the two algorithms for ten targets of the DUD-E benchmark dataset.

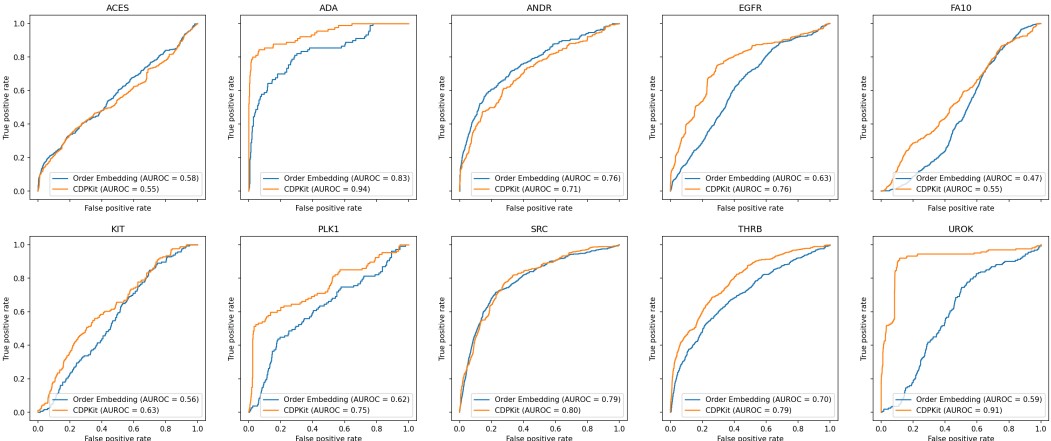

Figure 9: Absolute screening performance of PharmacoMatch and the alignment algorithm performance for ten targets of the DUD-E benchmark dataset. The pharmacophore queries were generated from the respective PDB ligand-receptor structures.

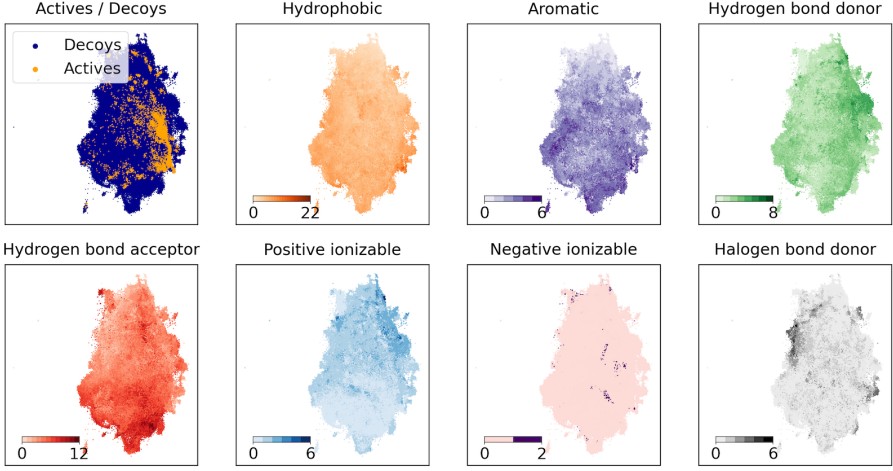

Figure 10: UMAP visualization of the vector embeddings of the ACES target.

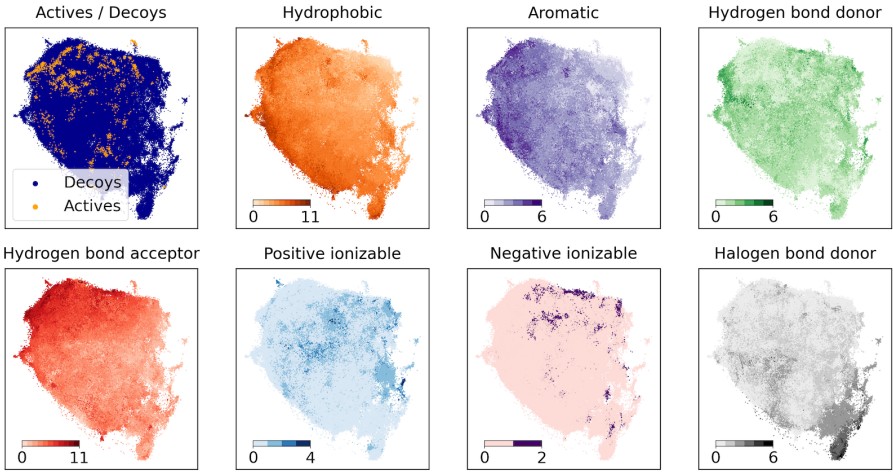

Figure 11: UMAP visualization of the vector embeddings of the ANDR target.

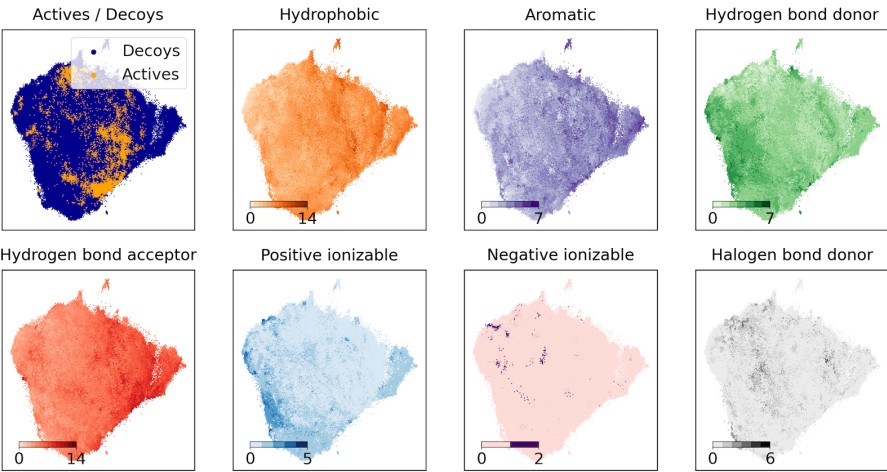

Figure 12: UMAP visualization of the vector embeddings of the EGFR target.

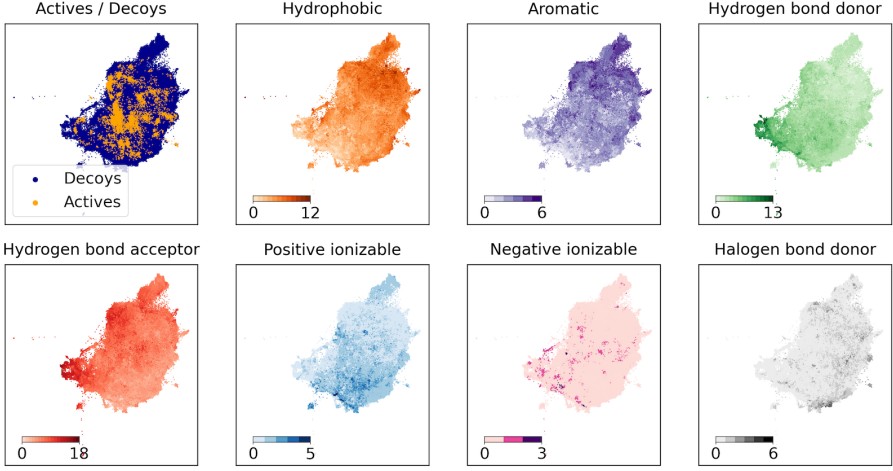

Figure 13: UMAP visualization of the vector embeddings of the FA10 target.

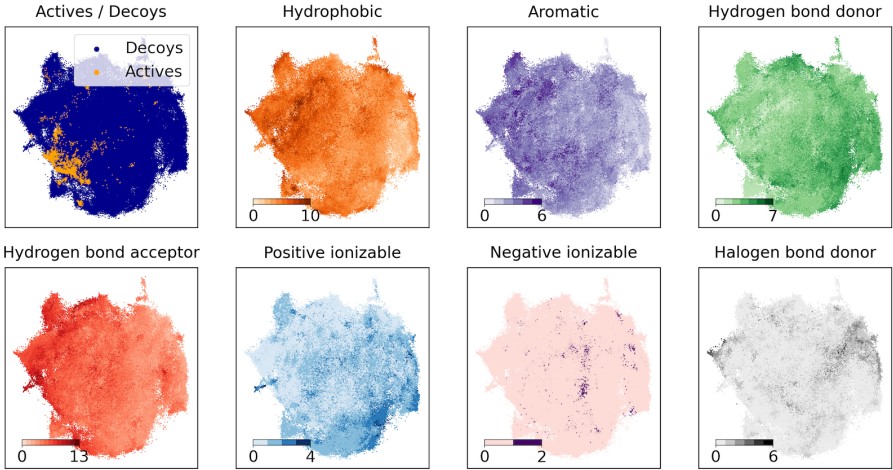

Figure 14: UMAP visualization of the vector embeddings of the KIT target.

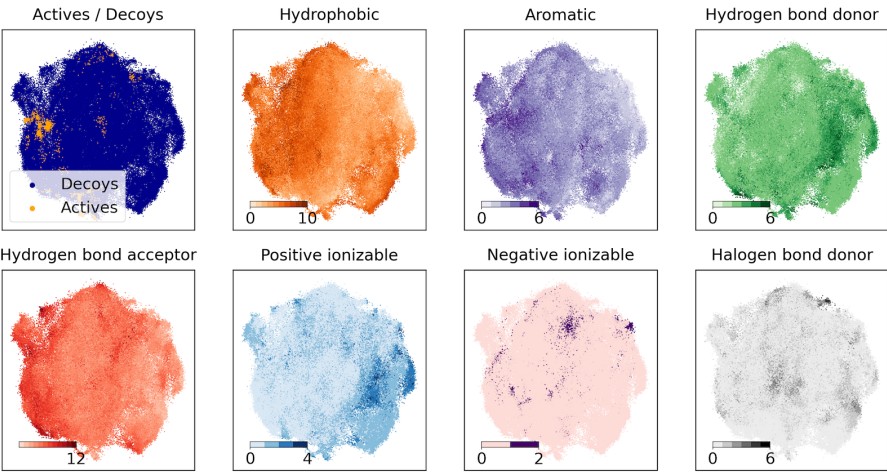

Figure 15: UMAP visualization of the vector embeddings of the PLK1 target.

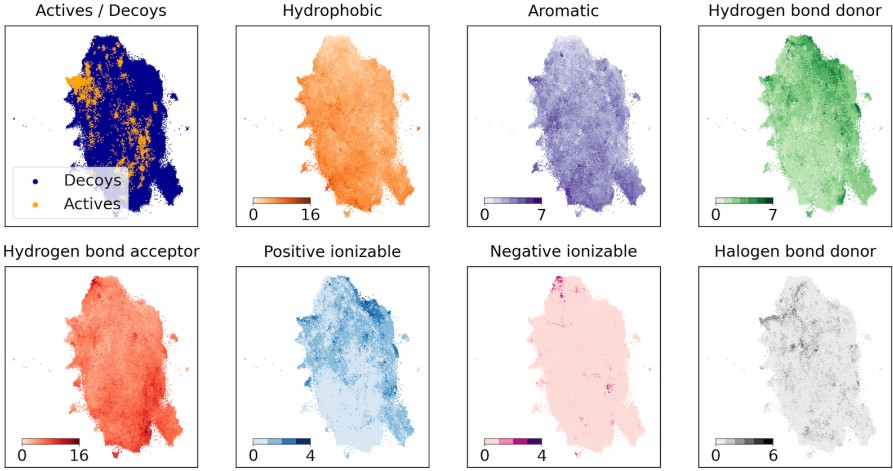

Figure 16: UMAP visualization of the vector embeddings of the SRC target.

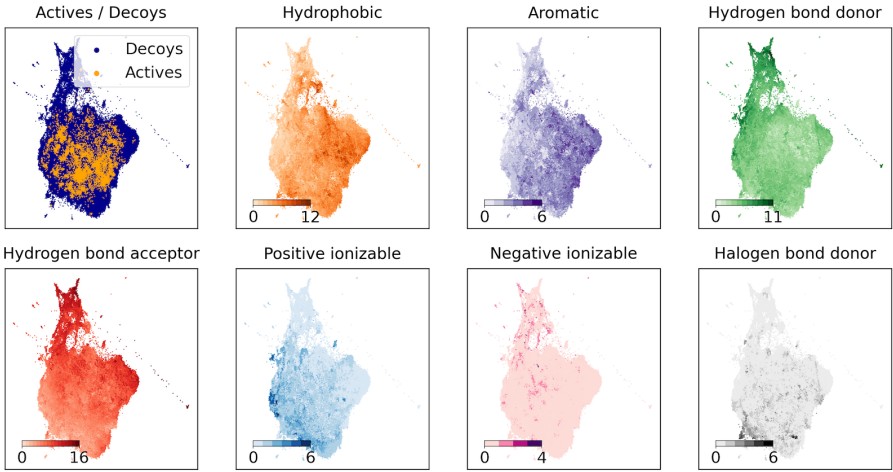

Figure 17: UMAP visualization of the vector embeddings of the THRB target.

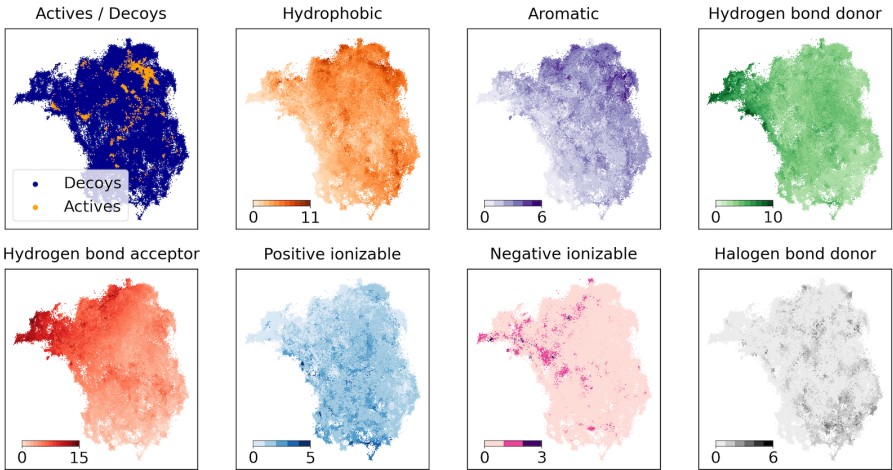

Figure 18: UMAP visualization of the vector embeddings of the UROK target.

