# OpenReview forum: "PharmacoMatch: Efficient 3D Pharmacophore Screening via Neural Subgraph Matching"
_ICLR.cc/2025/Conference — ICLR 2025 Poster_

### Official Review · Reviewer_2qzD · 2024-10-22

**Soundness:** 3
**Presentation:** 3
**Contribution:** 2
**Rating:** 5
**Confidence:** 3

**Summary:**

In this paper, the authors propose a contrastive learning approach for pharmacophore screening, based on subgraph matching. The key idea is to employ approximate subgraph matching for querying conformational database, a main step in pharmacophore screening. The subgraph matching is done through a contrastive learning approach by encoding query-target relationships in the embedding space. Their model has been validated based on benchmark dataset including DUD-E.

**Strengths:**

It is interesting to reinterpret pharmacophore screening problem as an approximate subgraph matching problem.

**Weaknesses:**

From the methodology point of view, the paper lacks novelty. The contrastive learning framework and augmentation module are all rather standard approach in GNN models. The contribution is not significant. Further, the performance is not very impressive as shown in Table 1. Even though the authors have emphasized that "our goal is to achieve comparable values between our model and the alignment algorithm", the only advantage of the current model seems to be the runtime efficiency.

**Questions:**

1) From Methodology point of view, is there any major novelty in the current paper?
2) Other than the efficiency in terms of the runtime, any other clear advantage of the current model?

---

> ### Author Response · Authors · 2024-11-22
>
> We would like to thank the reviewer for their valuable feedback.
>
> Q1: In our paper, we propose a learning algorithm for predicting the alignment of pharmacophores, to the best of our knowledge, an approach that has not been attempted previously. We believe that the insights from our work could be applied to other domains where the alignment of attributed point clouds is of interest. For example, point cloud matching is also a significant topic in the computer vision community, and we expect our framework could provide useful contributions to this area as well.
>
> Q2: Pharmacophore screening is a well-established technique that has been successfully used in the drug discovery community for over two decades. It serves as a low-cost prefiltering step to select a hitlist from a database, which is then followed by more computationally expensive methods such as Molecular Dynamics simulations. Although comparably cheap, the most significant bottleneck is the runtime, and we believe that improving this aspect will have the greatest impact in pharmacophore screening. Beyond virtual screening, we also believe our model could have broader applications. The learned embeddings from our model could serve as a 3D pharmacophore descriptor, opening up new possibilities. For instance, these embeddings could be used for clustering ligand-based pharmacophores of active compounds to create a shared feature pharmacophore, or they could facilitate the training of machine learning models for target-specific activity prediction.

---

> ### Comment · Area_Chair_MeUv · 2024-11-25
>
> Dear reviewer,
>
> Please at least acknowledge the rebuttal of the authors.
>
> Thanks,\
>  — Your AC

---

### Official Review · Reviewer_MX9K · 2024-10-28

**Soundness:** 2
**Presentation:** 2
**Contribution:** 1
**Rating:** 6
**Confidence:** 3

**Summary:**

The authors propose a contrastive learning approach that emphasizes augmentation strategies by incorporating the concept of pharmacophore in neural subgraph matching. Furthermore, they apply this concept to ligand-based virtual screening, demonstrating that the results are well-aligned with CDPKit’s alignment algorithm. Although the learned representations effectively capture the proposed pharmacophore concepts, the performance in virtual screening—a primary objective of the model—does not appear sufficiently strong. This is primarily due to the lack of benchmarking against other models and the use of additional evaluation metrics.

**Strengths:**

**S1. Alignment of Contribution and Results:**
	The study presents a coherent alignment between its contributions and the resulting outcomes. The objectives set forth by the authors are consistently addressed throughout the work.

**S2. Effective Representation Learning via contrastive learning:**
	The approach to representation learning through a contrastive learning with data augmentation appears to function as intended. The learned representations for pharmacophores are well-clustered in embedding space.

**Weaknesses:**

**W1. Limited Methodological Novelty:**
	The proposed methods do not introduce significant novel approaches. As mentioned in the manuscript, the concepts of Neural Subgraph Matching or neural network architectures are already proposed by other previous works. While the formulation applied to pharmacophores—particularly the augmentation strategies for contrastive learning—is noteworthy, it may not sufficiently advance the general methodologies handled in the ICLR community. If there's any other novelty compared to previous works, the points should be clearly explained in the manuscript. Authors may propose novel input processing schemes or model architectures better suited to the pharmacophore graph, or improve neural subgraph matching techniques.

**W2. Insufficient Experimental Results:**

- **W2.1 Lack of Comprehensive Benchmarks:**
	For a study proposing algorithms intended for virtual screening, it is essential to benchmark against established methods such as DrugClip or PharmacoNet to demonstrate comparative advantages. The absence of such comparisons, without a compelling justification, weakens the evaluation of the proposed method’s efficacy. Additionally, reliance on the outdated DUD-E benchmark and the arbitrary selection of only 10 targets out of 102 weakens the robustness of the experimental validation. (see Q4, 5)

- **W2.2 Ambiguous goal of benchmark experiment:**
	The goal of "*to achieve comparable values between our model and the alignment algorithm*" would be meaningful only if the alignment method inherently guarantees superior results compared to the previous methods, which is not adequately demonstrated in the current manuscript. (see Q5-1)

- **W2.3 Suitability for Virtual Screening:**
	The method appears to focus on ligand interactions with only parts of the protein pharmacophore, potentially neglecting the comprehensive information of the global protein binding site. This partial consideration might introduce bias, and it remains unclear whether the method can outperform existing techniques that utilize complete protein-ligand information. Empirical evidence showing superior performance in this regard would strengthen the study. (see Q6)

**Questions:**

## **Methods:**
**Q1. Clarification of Pharmacophore Representation:** In the section detailing pharmacophore representation, it is mentioned that $\mathcal{L}$ comprises only pharmacophoric descriptors. However, distances are subsequently incorporated into the representation. The notation and description should be revised for consistency and clarity.

**Q2. Model Input Specification:** Within the model input section, node labels are currently denoted as $V_p$, which is a set of node. It may be more appropriate to represent these as node element to enhance the clarity.

**Q3. Negative Data Augmentation Strategy:**
The current approach limits displacement directions to a single direction to avoid cancellation effects. Allowing for all displacement directions except for the case of cancellation can highly increase the diversity of negative training data, might lead to better model performance. Are there specific reasons why the authors chose to use only a single direction for negative displacements?

## **Results:**

**Q4. Evaluation on Recent Datasets:** To better assess the method’s applicability to real-world scenarios, evaluation on more recent datasets like LIT-PCBA[1] is recommended. Additionally, utilizing metrics beyond AUROC, such as enrichment factors (EF) (similar to BEDROC for early recognition of hit candidates), could provide a better understanding of the model’s performance in virtual screening tasks.

**Q5. Benchmarking with other methods:** Since one of the main contributions of this paper is "*fast virtual screening in the embedding space and evaluate the performance of our method through experiments on virtual screening benchmark datasets*", the authors should benchmark their virtual screening performance with other models. It seems that there are no significant differences in the objectives or methods of the compared models relative to the current work. Are there reasons why the authors did not benchmarked their model with the previous works such as DrugClip or PharmacoNet?

**Q5-1. Ambiguous criteria for showing virtual screening performance:**  The authors compared their performance with CDPKit's alignment algorithm. It only make sense that the propose method does well on virtual screening only if CDPKit alighment algorithm's performance is already good enough. Authors may provide the performance of CDPKit alignment algorithm for virtual screening explicitly in their manuscript.

**Q6: Limitation in Protein-Specific Virtual Screening Capability:**
The methodology seems similar to ligand-based approaches, where ligand structures are pre-generated and graph matching determines activity. This may limit the model's applicability, since considering only ligand structure of protein-ligand complex cannot fully consider protein pocket. For example, if protein has large binding site that various ligands with different pharmacophoric sites can interact with, considering only a single ligand might give a bias during virtual screening on the protein target. Why did authors adopted ligand-based approach instead of protein-based one, such as using protein binding site's pharmacophore instead its binding ligand?

## **References:**
[1] Tran-Nguyen, Viet-Khoa, Célien Jacquemard, and Didier Rognan. "LIT-PCBA: an unbiased data set for machine learning and virtual screening." Journal of chemical information and modeling 60.9 (2020): 4263-4273.

---

> ### Author Response · Authors · 2024-11-22
>
> We would like to thank the reviewer for their valuable feedback and address the questions raised:
>
> Q1: We kindly ask for clarification regarding the reported inconsistency. The label set $\mathcal{L}$ (pharmacophoric descriptors) indeed contains only the labels. However, the pharmacophore P consists of a set of tuples, where each tuple contains a pharmacophoric descriptor label $l_i$ and its corresponding 3D spatial location $r_i$. The pharmacophore graph is then constructed by assigning each node $v_i$ the descriptor label $l_i$, and edges $e_{ij}$​ represent the distances between points $i$ and $j$.
>
> Q2: We appreciate the reviewer’s detailed proofreading. We have revised the notation as suggested.
>
> Q3: The reason we chose this strategy is that random sampling of positions, without cancellation, does not ensure avoidance of cases where a negative pair augmentation strategy could accidentally create a positive pair. To address this issue, we implemented the current strategy, which has shown to outperform random sampling in terms of model performance.
>
> Q4 & Q5: Pharmacophore screening is inherently an interactive process, where users design queries to test on a validation set of known active and inactive compounds. Since the query design significantly influences the outcome, the results of both the CDPKit algorithm and PharmacoMatch are dependent on the chosen query.
> To capture different aspects of the evaluation, we report both absolute and relative performances of our method. Absolute performance provides an overview of the query-dependent virtual screening performance of both the CDPKit algorithm and our trained model. Relative performance, on the other hand, reflects how well our model approximates the outcome of the CDPKit alignment, which is the primary focus of our work. As suggested by the reviewer, we have added the enrichment factor metric in Table 1 of the revised manuscript.
> Regarding the use of the DUD-E dataset, we believe that testing on all 102 targets would not offer significant additional value, as the results are query-dependent. Therefore, we chose to illustrate our findings using a subset of 10 targets, which we consider a more focused case study. The query dependence also prevents direct comparisons with other methods on standard benchmarks. For example, DrugCLIP works in an automated manner using the complete receptor structure as a query, while PharmacoNet generates an automated query via image segmentation. Neither of these methods incorporates user interaction, which is a central component of pharmacophore screening. Our contribution is in providing a faster method that supports user interaction by enabling more efficient hit retrieval and reducing runtimes, ultimately streamlining the workflow.
>
> Q5-1: Since our method predicts the outcome of an alignment, it is necessary to compare it with an alignment algorithm. As noted by the reviewer, our best model can only perform as well as the traditional alignment algorithm. We do explicitly report the alignment results of the CDPKit algorithm in Table 1, specifically in the "absolute screening performance" column.
> At this point, we would like to highlight the broader value of pharmacophore screening in the drug discovery pipeline. The goal of pharmacophore screening is not to develop the best-performing classifier. The information content of a pharmacophore query is typically too limited to yield optimal results for activity prediction. Rather, pharmacophore screening serves as an efficient prefilter, focusing on initial enrichment of a hitlist, which can then be further refined using more computationally expensive methods in subsequent studies.
>
> Q6: We define structure-based pharmacophores as those derived from protein-bound ligands, as described in [1]. While this approach does not capture all binding hotspots within the protein pocket, it is a standard practice in structure-based pharmacophore modeling. A ligand-based approach would involve designing the pharmacophore query based solely on known active and inactive compounds, without a receptor-ligand structure. We would like to emphasize that our method is versatile and can be applied in both cases, as our model approximates the alignment algorithm and is not dependent on a specific target structure.
>
> We hope these clarifications address the reviewer’s concerns and improve the clarity of our work. Once again, thank you for your thoughtful and constructive feedback.
>
> [1] Gerhard Wolber and Thierry Langer. Ligandscout: 3-d pharmacophores derived from protein-bound ligands and their use as virtual screening filters. J. Chem. Inf. Model, 45(1):160–169, 2005.

---

> > ### Comment · Reviewer_MX9K · 2024-11-23
> >
> > **[Reply on Q1]** I apologize for my earlier question being unclear. To clarify, the part I found confusing was the definition of $\lambda$ as $V \times E \to \mathcal{L}$. While I understand how $V \to \mathcal{L}$ can be defined as stated, it’s unclear how ${E} \to \mathcal{L}$ is defined if $\mathcal{L}$ does not include distance. If I have misunderstood something, please feel free to correct me.
> >
> > While this detail may not significantly impact the overall contribution of the work, I believe it could improve the readability of the manuscript. Including distance in $\mathcal{L}$ or defining a separate set specifically for distance could make this aspect clearer. However, I understand that the latter option might require more substantial revisions. Thank you for considering this suggestion.
> >
> > **[Reply on Q2]** Thanks for the authors revision.
> >
> > **[Reply on Q3]** The reason is acceptable. However, I think the reason in the reply should be included in the manuscript for better understanding about negative data augmentation strategies.
> >
> > **[Reply on Q4, 5]**
> > - [About Contribution] After reviewing your response, I revisited the main contributions highlighted in your work. The key contributions of PharmacoMatch are: (1) learning pharmacophore representations using GNNs and (2) enabling fast virtual screening. However, I believe it is necessary to clearly distinguish between virtual screening (VS) and pre-screening, and explicitly state that PharmacoMatch focuses on the latter. There are many models for virtual screening, including DrugClip, and claiming efficiency in VS without comparing against such models may not provide strong supporting evidence for this point.
> >
> > - [Comparison with other works] While I agree that user interaction is an important advantage of this work, I still think that the current result is not sufficient to demonstrate the effectiveness of PharmacoMatch in practical tasks, especially in the pre-screening task. To highlight the practical strengths of this approach, it would be essential to clearly show the effectiveness of the representation learning and scoring methods. Regarding the earlier question about a comparison with PharmacoNet, I believe the response that comparison with PharmacoNet is not available due to unavailability of user-interaction does not fully justify the absence of a comparison. My understanding is that, PharmacoNet, **whose main goal appears to be pre-screening like PharmacoMatch**, first generates pharmacophore queries via segmentation and then uses a non-deep learning-based method for scoring. Given that one of the core contributions of PharmacoMatch is its ability to effectively learn representations of pharmacophore graphs and use them for pre-screening, I think **authors can strongly demonstrate PharmacoMatch's performance on practical tasks through their contributions by showing that PharmacoMatch performs better when pre-screening pharmacophore queries generated by PharmacoNet’s segmentation method.** This would more directly highlight the strength of the proposed representation learning approach and its practical applicability, possibly more effectively than the current partial pre-screening experiment (Experiment 3) on DUD-E. If there's any ambiguity in my question, please let me know.
> >
> > - [Detailed experiment settings] In the Appendix, it is mentioned that pharmacophore queries were obtained from respective PDB ligand-receptor structures. While I assume this involves extracting ligand-bound receptor regions, the manuscript currently lacks a detailed explanation of the exact procedure. Providing a more explicit description of how these pharmacophore queries were generated would improve clarity and enhance the reproducibility of the work.
> >
> > **[Reply on Q6]** Thank you for explanation.
> >
> > **[Reply on General response]**
> > Regarding reproducibility and dataset usability, it is difficult to assess these aspects without access to publicly available code and data.
> >
> > Thanks to the authors for their kind rebuttal. However, I strongly think that if the manuscript includes a comparison with PharmacoNet as suggested earlier, it would significantly strengthen the evaluation and highlight the practical benefits of PharmacoMatch. If the authors incorporate such results, I will adjust my score from 5 to 6. Once again, if there's any ambiguity in my response, feel free to reply on this comment.

---

> > > ### Author Response · Authors · 2024-11-23
> > >
> > > We sincerely thank the reviewer for their prompt and detailed feedback on our rebuttal. We will try to incorporate the suggested improvements before the discussion phase ends. Regarding reproducibility and dataset usability, we would like to emphasize that our reproducibility statement includes an anonymized link to both our source code and the processed data used in our experiments.

---

> ### Author Response · Authors · 2024-11-26
>
> We would like again to thank the reviewer for their insightful questions.
>
> Q1:
> Thank you for your clarification regarding concerns about the inconsistency in our notation. We now understand the source of confusion and have redefined the set of labels, $\mathcal{L} = \mathcal{D} \cup \mathcal{R}$, as the union of the set of descriptor types $\mathcal{D}$ and the set of pairwise distances $\mathcal{R}$. We hope this revision enhances the readability and consistency of the manuscript.
>
> In your question, you referred to the labeling function $\lambda : V \times E \rightarrow \mathcal{L}$, defined as operating on the Cartesian product of the vertex set $V$ and the edge set $E$. To avoid further confusion, we wish to emphasize that our definition of $\lambda : V \cup E \rightarrow \mathcal{L}$ is based on the union of these sets. This notation was adapted from the ICML publication [1], which we have cited in the paper.
>
> [1] Nils Kriege and Petra Mutzel. Subgraph matching kernels for attributed graphs. In Proceedings of the 29th International Conference on Machine Learning, pp. 291–298, 2012.
>
> Q3:
> We have added our reasoning to the description of the augmentations in Section 4.
>
> Q4,5:
> To highlight our conceptual contribution, we have emphasized this aspect by adding “prediction of pharmacophore matching using learned representations” to our contributions section in the introduction. Recognizing that our method's primary impact lies in its utility as a prescreening tool, we have revised this point across the paper, including the contributions, abstract, and other relevant sections.
>
> To accommodate these revisions while adhering to the 10-page limit, we removed the introduction on SSL from the related works section and relocated the workflow illustration overview to the supplementary information. Additionally, we have clarified the role of user interaction in query refinement in Section 5.3.
>
> We also added an experiment comparing the prescreening capabilities of our method with PharmacoNet. Below, we outline the design choices and challenges that guided our experimental procedure:
>
> Use of LIT-PCBA:
> We did not test on the LIT-PCBA benchmark. The authors of PharmacoNet provide only averaged metrics across all targets in the dataset. LIT-PCBA includes multiple receptor structures for each target. Since the authors do not specify which structures were used to generate queries, and also do not provide preprocessed data, we were unable to perform this comparison.
>
> Use of DEKOIS2.0:
> The DEKOIS2.0 benchmark was more suitable, as it includes only one receptor structure per target. However, query generation remained ambiguous. In cases where the small-molecule binder was present in multiple binding pockets, the authors did not specify which ligand was used. We randomly selected one binding pocket for pharmacophore generation.
>
> Query Generation:
> We did not use queries generated by PharmacoNet, as suggested by the reviewer. Pharmacophores generated by different software platforms have been shown to be incompatible, and we have cited relevant literature to support this claim [2]. Instead, we used structure-based queries generated by CDPKit without any refinement, simulating an automated workflow without user intervention. These queries were then used for prescreening and compared with PharmacoNet.
>
> Conformer Differences:
> The conformers used by CDPKit and PharmacoNet differ (CDPKit uses its own conformer generation, while PharmacoNet relies on RDKit). Unfortunately, the authors of PharmacoNet did not include their processed data in the supplementary information, making it unclear how their conformers were generated. This introduces some uncertainty into the comparison.
>
> Taking these factors into account, we compared the averaged prescreening performance metrics of PharmacoMatch and PharmacoNet. Our results show that while PharmacoMatch exhibits a slight decrease in enrichment metrics, it achieves a 1000-fold faster prescreening time compared to PharmacoNet. We believe this represents a substantial improvement in the context of prescreening large datasets.
>
> We sincerely thank the reviewer for their insightful questions, which have led to significant improvements in our manuscript. We hope that the additional experiments and revisions convincingly demonstrate the value and impact of our method.
>
> [2] Spitzer et al., J. Chem. Inf. Model. 2010, 50, 7, 1241–1247

---

> > ### Comment · Reviewer_MX9K · 2024-11-27
> >
> > Thanks to the authors for kind response and conducting the experiments that I've suggested.
> >
> > The additional experiment is acceptable given PharmacoMatch's primary goal is pre-screening.
> >
> > I have minor suggestions for the manuscript. (Regarding questions 4, 5) Is there a reason to separate "Prediction of pharmacophore matching using learned representations" with the last contribution? I think the first contribution alone might be ambiguous.
> >
> > Although I think more benchmark results as in additional experiment are needed to prove the "practical" performance of PharmacoMatch, I have adjusted my score to 6 as most of my concerns are resolved.
> >
> > Thanks again to the authors for their response.

---

> > > ### Author Response · Authors · 2024-11-27
> > >
> > > Thanks for pointing that out. The decision to separate the contributions was intended to highlight three distinct aspects: a conceptual contribution, a methodological one, and a practical one. Based on your feedback, we have revisited and rephrased the contributions section to ensure it better reflects this structure. We hope the revised version addresses your concern.
> > >
> > > Thanks again to the reviewer for their valuable input.

---

### Official Review · Reviewer_p5tV · 2024-11-02

**Soundness:** 2
**Presentation:** 2
**Contribution:** 2
**Rating:** 5
**Confidence:** 4

**Summary:**

The authors introduce PharmacoMatch, a deep learning approach that reframes 3D pharmacophore screening as a neural subgraph matching problem, using a graph neural network trained through contrastive learning to encode pharmacophore structures into an embedding space. Their method achieves comparable screening performance to traditional alignment-based approaches while being approximately two orders of magnitude faster in matching speed, making it particularly valuable for screening extremely large molecular databases. The model is trained in a self-supervised manner on over 1.2 million unlabeled molecules from ChEMBL, learns to encode both structural and spatial relationships of pharmacophoric points, and demonstrates robust performance across multiple DUD-E benchmark datasets in a zero-shot setting.

**Strengths:**

The acceleration is impressive, with a thorough evaluation of embeddings and runtime analysis. The model provides a practical impact for screening billion-compound libraries. Besides, the reformulation of pharmacophore screening as neural subgraph matching is creative combining self-supervised training approach using augmentation strategies.

**Weaknesses:**

Despite mentioning recent works like DrugClip and PharmacoNet in the related work section, there are no direct comparisons with these methods. It seems the paper only compares against the traditional CDPKit alignment algorithm, missing comparisons with other more current learning approaches. Besides, there is no comparison or detailed discussion with simpler baseline models (e.g., basic GNN architectures without contrastive learning)

The discussion on the model details is not sufficient. Lacks systematic ablation studies of model architecture components (e.g., the impact of different GNN layers, the importance of skip connections). Missing analysis of the impact of different augmentation strategies on model performance and investigation of how embedding dimension and model size affect performance, as well as contrastive loss function impacts.

**Questions:**

1. How sensitive is the model to the choice of tolerance radius (r_T = 1.5Å)? Could you provide an analysis?
2. How did you select the 10 DUD-E targets? Could you demonstrate the method's robustness on more targets?
3. What is the impact of different augmentation strategies? For example, how much does performance degrade if you remove node deletion or displacement?
4. Could you compare PharmacoMatch with recent methods like DrugClip and PharmacoNet on the same benchmarks?
5. The speed advantage is clear, but can the author better justify the conceptual novelty? (Contrastive learning is not new, GNN for molecules is well-established and order embeddings have been used before)
6. How much 3D geometric precision is actually lost during embedding? Could you quantify the tradeoff between speed and accuracy?
I wonder if there are specific types of 3D arrangements where the method may fail.

---

> ### Author Response · Authors · 2024-11-22
>
> Thank you for reviewing our work and for acknowledging both the runtime improvements and the creativity of the proposed framework. We would like to address the reviewer’s questions and concerns in detail:
>
> 1. Radius of 1.5: The choice of a radius of 1.5 is based on the default radius of the CDPKit alignment algorithm. Augmenting samples with the same radius ensures consistency and achieves the best performance. We have added ablation studies with different radii, which confirm that 1.5 works best in this setting.
>
> 2. Selection of DUD-E targets: The 10 DUD-E targets were selected randomly. As noted in our response to shared concerns, adding more benchmark experiments would not be particularly informative in our unique setting. Pharmacophore screening fundamentally depends on query design, making direct comparisons with standardized benchmarks difficult. The key metric, in our opinion, is the relative comparison to the existing alignment algorithm, which we aim to approximate. While absolute virtual screening metrics are valuable, we present them as a case study to showcase the application of our framework.
>
> 3. Node deletion and displacement: Node deletion is essential for our approach. It enables the creation of subsets from the batched pharmacophore graphs, which are then used as queries in the subgraph matching task. Without valid subgraphs, training a model for subgraph matching would not be feasible. On the other hand, the question on the importance of node displacement is valid. We have included experiments on this in our ablation studies and can confirm that removing node displacement (displacement radius = 0 Angstrom) degrades model performance.
>
> 4. Comparison to DrugCLIP and PharmacoNet: Due to the unique nature of pharmacophore screening, direct comparisons with methods like DrugCLIP or PharmacoNet are not feasible. Our method's reliance on query design makes benchmarking against other approaches difficult. DrugCLIP operates automatically by using the complete receptor structure as a query, while PharmacoNet generates an automated query through image segmentation for screening. Both methods do not incorporate user interaction, which is a key component of our pharmacophore screening process.
>
> 5. Novelty of the framework: While the individual components of our model are not novel, their combination is. Our method predicts pharmacophore alignment using learned representations, which, to the best of our knowledge, has not been attempted before. For comparison, DrugCLIP (NeurIPS 2023) applies the well-known CLIP framework and UniMol encoder to protein and molecular data, with its novelty lying in adapting these tools to a new problem, supported by a customized training strategy and augmentations. Similarly, our approach makes a significant contribution by addressing a real-world challenge through a novel integration of established methods. We believe this contribution is both meaningful and impactful. As emphasized in the ICLR guidelines, contributions in applied domains are actively encouraged.
>
> 6. Geometric precision: We kindly request the reviewer to clarify what they mean by geometric precision. In our evaluation (Section 5, “Screening Performance” & “Runtime comparison”), we quantify the speed/accuracy trade-off. Specifically, the “relative performance” column in Table 1 reflects the correlation between our model's alignment predictions and those of the CDPKit algorithm. Limiting 3D arrangements: We acknowledge that our model can fail in certain cases. For example, the E(3)-invariant encoder cannot distinguish a pharmacophore from its mirror image, potentially increasing the false positive rate. Addressing this limitation with an SE(3)-invariant model will be part of future work.
>
> Further questions:
>
> Basic GNN without contrastive learning: Using a basic GNN without contrastive learning would not align with the objectives of our study. Our goal is to model the prediction of an alignment algorithm, which inherently requires a function with two inputs. A GNN could be used for activity prediction, but this approach would not be relevant in our context.
>
> Ablation studies: In the revised manuscript, we have included detailed ablation studies to address the reviewer’s questions about the model components (Section 4, “Ablation studies”).

---

> ### Comment · Area_Chair_MeUv · 2024-11-25
>
> Dear reviewer,
>
> Please acknowledge the rebuttal of the authors.
>
> Thanks!
> — Your AC

---

> > ### Comment · Reviewer_p5tV · 2024-11-26
> >
> > I appreciate the author's effort in answering the questions above. I have adjusted the score since the author mentioned they will add ablation studies to make the work more solid and comprehensive.

---

> > > ### Author Response · Authors · 2024-11-27
> > >
> > > We sincerely thank the reviewer for acknowledging our ablation studies. We are pleased to inform the reviewer that we have updated the manuscript with an additional study in Section 5.3 to address Question 4. Specifically, we compare our method, PharmacoMatch, with PharmacoNet on the DEKOIS2.0 benchmark.
> > >
> > > In this experiment, we evaluate the average prescreening performance metrics of PharmacoMatch and PharmacoNet. The results indicate that while PharmacoMatch exhibits a marginal decrease in enrichment metrics, it achieves a 1000-fold improvement in prescreening speed compared to PharmacoNet. This significant acceleration highlights the practicality of PharmacoMatch for prescreening large datasets, where computational efficiency is often a critical factor.
> > >
> > > Further details and additional explanations regarding this experiment are provided in our discussion with Reviewer MX9K. We hope these enhancements and the added context substantiate the substantial value and real-world applicability of our approach.

---

### Official Review · Reviewer_GwAp · 2024-11-07

**Soundness:** 3
**Presentation:** 3
**Contribution:** 3
**Rating:** 6
**Confidence:** 4

**Summary:**

This paper introduces a novel contrastive learning approach based on neural subgraph matching, i.e., PharmacoMatch, and the authors claim that it reinterprets pharmacophore screening as an approximate subgraph matching problem and enables efficient querying of conformational databases by encoding query-target relationships in the embedding space.

**Strengths:**

+ This paper is well-written and nicely organized.
+ The proposed framework is novel and is nicely motivated.

**Weaknesses:**

- I suggested the authors consider conducting more experiments over other datasets instead of only DUD-E.
- I wonder if the proposed contrastive learning approach can be applied to other domain datasets?
- This paper does not provide any unique and novel insights about why the proposed architecture that works well for the current dataset.

**Questions:**

See comments in the Weaknesses part.

---

> ### Author Response · Authors · 2024-11-22
>
> Thank you very much for reviewing our paper and for your kind words about its novelty and the proposed framework.
>
> 1. As mentioned in our official response to shared concerns, we believe that adding more benchmark experiments in our unique setting would not be particularly informative. Pharmacophore screening fundamentally relies on query design, which makes direct comparisons with standardized benchmarks challenging. In our view, the most important metric is the relative comparison to an existing alignment algorithm, as this is the method we aim to approximate. While absolute virtual screening metrics are also important, we present them primarily as a case study to illustrate the application of our framework.
>
> 2. Our work demonstrates how the proposed framework can be applied to 3D geometric graphs. Neural subgraph matching has been utilized in diverse domains, such as molecular graph matching and word embeddings. We believe our contribution to attributed point clouds could be relevant to applications like matching of point clouds from LIDAR scans, which should broaden the framework's impact.
>
> 3. We would kindly ask the reviewer to clarify their question, as its current phrasing could be interpreted ambiguously.  If the inquiry pertains to ablation studies, we have added a paragraph in the methodology section to address this.

---

> > ### Comment · Reviewer_2qzD · 2024-11-25
> >
> > The Reviewer MX9K does a great job of listing all the detailed concerns, which I totally agree with. In particular, the concern of "PharmacoMatch's performance on practical tasks". I will still keep my score.

---

> > > ### Author Response · Authors · 2024-11-26
> > >
> > > We kindly request the reviewer to revisit the updated version of our paper, where we have added experiments in Section 5.3 showcasing the performance of PharmacoMatch on a practical task. Specifically, we compare it with PharmacoNet on the DEKOIS2.0 benchmark. We hope these results demonstrate the value of our contribution.

---

> > ### Comment · Reviewer_GwAp · 2024-12-03
> > **Thanks for your response**
> >
> > I would like to thank the authors for their responses and the updated revision. Ablation studies is helpful. Somehow I still think adding more benchmark will help illustrate the power of proposed PharmacoMatch algorithm cross-domain. I changed my score from 5 to 6 accordingly.

---

> ### Comment · Area_Chair_MeUv · 2024-11-25
>
> Dear reviewer,
>
> Please acknowledge the rebuttal of the authors.
>
> Thanks!\
> — Your AC

---

### Author Response · Authors · 2024-11-22

We sincerely thank the reviewers for their thoughtful feedback and the time they dedicated to evaluating our work. We address the shared concerns raised, beginning with the conceptual novelty of our approach.

Our paper presents a method to predict the alignment of pharmacophores using a learning-based algorithm, which, to the best of our knowledge, is the first of its kind. While the primary focus is on applications in virtual screening for drug discovery, we believe that our insights extend beyond this domain. They could be valuable in other areas where the alignment of attributed point clouds is of interest, such as point cloud matching in computer vision.

Although the individual components of our model are not entirely novel, their unique combination addresses a previously unexplored problem. Contributions like ours are significant as they showcase how existing methodologies can be creatively adapted to solve real-world challenges. For instance, the DrugCLIP framework (NeurIPS 2023), referenced by the reviewers, leverages the established CLIP framework and the Unimol encoder to apply them to molecular and protein data. Similarly, our work combines carefully selected building blocks, training strategies, and custom augmentations to tackle pharmacophore alignment effectively. This application-driven contribution aligns well with ICLR's encouragement of submissions addressing interdisciplinary challenges in fields like chemistry and drug discovery.

Another key contribution of our work is its emphasis on reproducibility. During the course of this project, we observed that many existing papers in the field lack crucial details about the design choices necessary for successful model training. To address this, we have provided a comprehensive description of our model architecture, and we commit to releasing both our code and the processed dataset. Additionally, our dataset represents a significant contribution on its own. It comprises tuples of 3D conformations and their corresponding pharmacophores, offering a valuable resource for the community to train and evaluate machine learning models in this domain.
In response to the feedback, we have enhanced the manuscript by adding an ablations paragraph to Section 4 (Methodology) and reporting the enrichment factor metric for all target datasets in Table 1.

However, we have not included additional benchmarks, and we did not compare to other frameworks such as DrugCLIP or PharmacoMatch, as pharmacophore screening operates fundamentally different. Its outcome depends not only on the dataset but also critically on the query design, which incorporates the expertise of medicinal chemists. This iterative process of query design, screening, and hitlist evaluation forms a feedback loop that cannot be automated, setting pharmacophore screening apart from other machine learning-based solutions.
While this approach benefits from the integration of expert knowledge, it complicates direct comparisons using standardized benchmarks. Simply running our method on predefined benchmarks and comparing metrics reported by other groups would not accurately capture its effectiveness. Instead, we report relative comparisons to an existing alignment algorithm, which better reflects the unique context and strengths of our work.

---

> ### Author Response · Authors · 2024-11-27
> **Comment on the Last Revision Update**
>
> In our most recent paper revision, we introduced a comparison between our method, PharmacoMatch, and an existing prescreening solution, PharmacoNet, using the established DEKOIS2.0 benchmark.
> This experiment evaluates the average pre-screening performance metrics of PharmacoMatch and PharmacoNet, employing structure-based pharmacophore queries without user refinement. We report the averaged screening performance across all targets in the benchmark. The results show that while PharmacoMatch exhibits a marginal decrease in enrichment metrics, it achieves a 1000-fold improvement in prescreening speed compared to PharmacoNet. This significant acceleration underscores the practicality of PharmacoMatch for prescreening large datasets, where computational efficiency is paramount.
> We would like to thank Reviewer MX9K for the insightful discussion, which directly inspired this valuable additional study. We believe these enhancements and the inclusion of this new benchmark strengthen the evaluation of our approach and further highlight its practical applicability.

---

### Meta-Review · Area_Chair_MeUv · 2024-12-20

**Metareview:**

This paper introduces a new virtual screening method based on contrastive learning via neural subgraph matching. Of particular relevance for me is that the method is substantially _faster_ than existing methods; this is an aspect of machine-learning models that we typically do not discuss sufficiently enough because we (wrongly) assume that faster GPUs are going to fix everything. Hence, this method has the potential to be also impactful in practical applications. Moreover, the authors invested substantial efforts in comparing and evaluating their method. This is exemplified, among other things, by the fact that the authors provide well-documented source code with easy-to-follow scripts for reproducing their results—this is certainly laudable and I wish more papers would do the same!

The strengths of the papers are very slightly marred by some minor issues, which are mostly concerning the selection of additional datasets and the question whether, beyond any computational performance gains, the results are substantially better than state-of-the-art methods. However, the rebuttal clearly indicated that the authors are more than willing to improve the evaluation of their method. In addition, while some reviewers (see below) had some concerns about the methodological contribution of the work, it is very much apparent to me that (a) the method is a creative combination of existing techniques (a sentence that applies to almost anything if viewed in the right light) and (b) this combination is leading to novel insights into a highly-relevant problem. As such, I agree with the authors, who mention that such interdisciplinary papers might often have a harder time being accepted in such a conference.

Given that during the rebuttal some reviewers were unfortunately not active, I am exercising my privileges as an area chair and suggest _accepting_ the work for presentation at the conference, since I believe the concerns by reviewers to have been sufficiently addressed by the rebuttal (see below for more details on this). I trust the authors to update their manuscript to alleviate some of the concerns readers might have, for instance concerning the evaluation/benchmark questions raised by some reviewers.

**Additional Comments On Reviewer Discussion:**

Reviewers agreed on the relevance of the work and appreciated the solution. Some minor weaknesses were raised concerning the evaluation and choice of comparison partners, as well as some methodological concerns (`GwAp`). The authors addressed these concerns to the reviewer's satisfaction. Reviewer `p5tV` initially raised questions about the methodology, which have been sufficiently addressed by the authors—I **strongly recommend** authors to modify their manuscript such that these questions are directly addressed by the text. Reviewer `MX9K` raised some concerns about reproducibility, which have been directly addressed by the authors through code and additional explanations. Finally, reviewer `2qzD` raised concerns about the novelty of the paper; the authors addressed this in their response but the reviewer did not further engage in the discussion or acknowledge the rebuttal. It is within my purview as an area chair to evaluate the rebuttal and as such, I believe these concerns to be sufficiently addressed. For their revision, the authors could add a brief paragraph reflecting on the _relevance_ of their contributions; I believe that a good place for this would be directly after mentioning their key contributions in the introduction.

---

> ### Public Comment · ~Daniel_Rose1 · 2025-02-13
>
> We sincerely appreciate your positive assessment. Below, we summarize the changes made to the camera-ready version:
> - We shortened the introduction for improved clarity.
> - As suggested, we added a paragraph on the relevance of our contributions.
> - We incorporated our responses to reviewer p5tV’s questions into the main text.
> - We redesigned and merged the model training and augmentation figures. This allowed us to reintroduce the overview figure.
> - We moved the paragraphs on ablation studies and curriculum training to the Appendix, with cross-references in the main text.
> - We added a formal description of the performance metrics used in the Appendix.
> - As suggested by the reviewers, we extended the pre-screening experiment to the LIT-PCBA dataset, where we report slightly higher enrichment than PharmacoNet.
> - We included descriptions of the DEKOIS2.0 and LIT-PCBA datasets in the benchmark datasets section.
> - We updated the links to our repository and preprocessed datasets, which are now publicly available.
> - Additionally, after the rebuttal phase, CDPKit (which provides the reference solution for our project) reported a bug fix related to the alignment algorithm and the placement of hydrophobic features (see the CDPKit homepage and the release notes for versions v1.2.0 & v1.2.1). To maintain reproducibility, we reran our preprocessing pipeline and recalculated performance metrics accordingly. The updates led to slightly improved relative performance metrics in the DUD-E experiment and improved pre-screening results on DEKOIS2.0, where we now outperform PharmacoNet in early enrichment metrics.
>
> We hope that these revisions comprehensively address the reviewers' suggestions. Once again, we thank the area chair and all reviewers for their valuable feedback, which has significantly strengthened our manuscript.

---

### Decision · Program_Chairs · 2025-01-22

Accept (Poster)